# DisEnvisioner: Disentangled and Enriched Visual Prompt for Customized Image Generation

**Jing He[1]\* Haodong Li[1]\* Yongzhe Hu Guibao Shen[1] Yingjie Cai[3] Weichao Qiu[3] Yingcong Chen[1,2] ✉**
[1]HKUST(GZ) [2]HKUST [3]Noah's Ark Lab
{jhe812, hli736}@connect.hkust-gz.edu.cn; yingcongchen@ust.hk

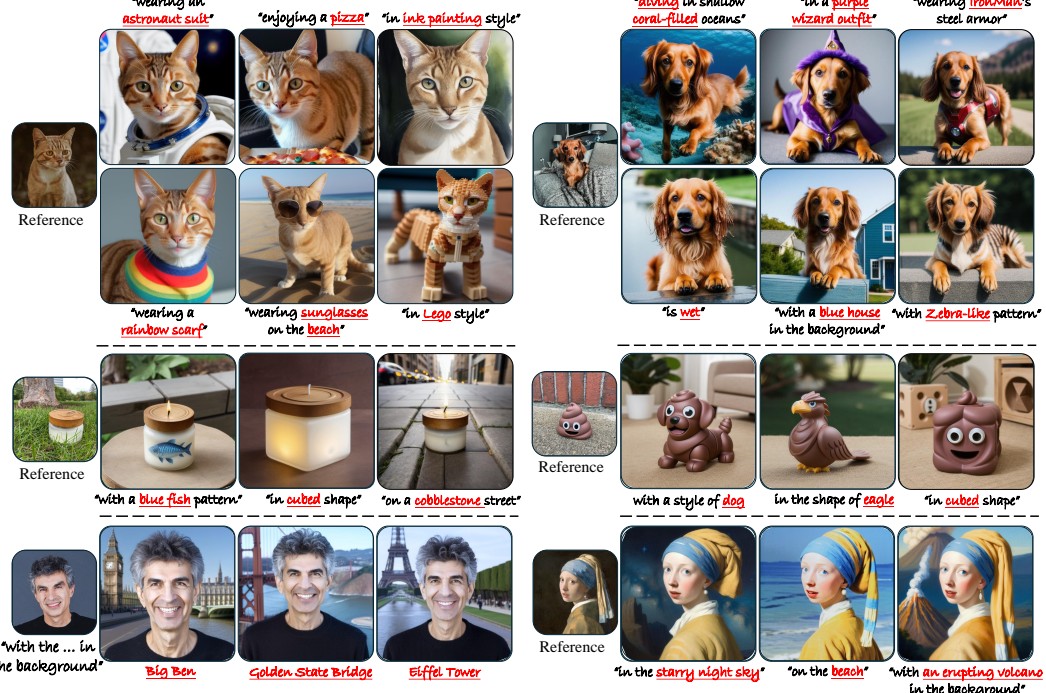

Figure 1: **Customization examples of DisEnvisioner.** Without cumbersome tuning or relying on multiple reference images, DisEnvisioner is capable of generating a variety of exceptional customized images. Characterized by its emphasis on the interpretation of subject-essential attributes, DisEnvisioner effectively discerns and enhances the subject-essential feature while filtering out irrelevant attributes, achieving superior personalizing quality in both editability and ID consistency.

## ABSTRACT

In the realm of image generation, creating customized images from visual prompt with additional textual instruction emerges as a promising endeavor. However, existing methods, both tuning-based and tuning-free, struggle with interpreting the subject-essential attributes from the visual prompt. This leads to subject-irrelevant attributes infiltrating the generation process, ultimately compromising the personalization quality in both editability and ID preservation. In this paper, we present **DisEnvisioner**, a novel approach for effectively extracting and enriching the subject-essential features while filtering out -irrelevant information, enabling exceptional customization performance, in a **tuning-free** manner and using only **a single image**. Specifically, the feature of the subject and other irrelevant components are effectively separated into distinctive visual tokens, enabling a much more accurate customization. Aiming to further improving the ID consistency, we enrich the disentangled features, sculpting them into a more granular representation. Experiments demonstrate the superiority of our approach over existing methods in instruction response (editability), ID consistency, inference speed, and the overall image quality, highlighting the effectiveness and efficiency of DisEnvisioner.

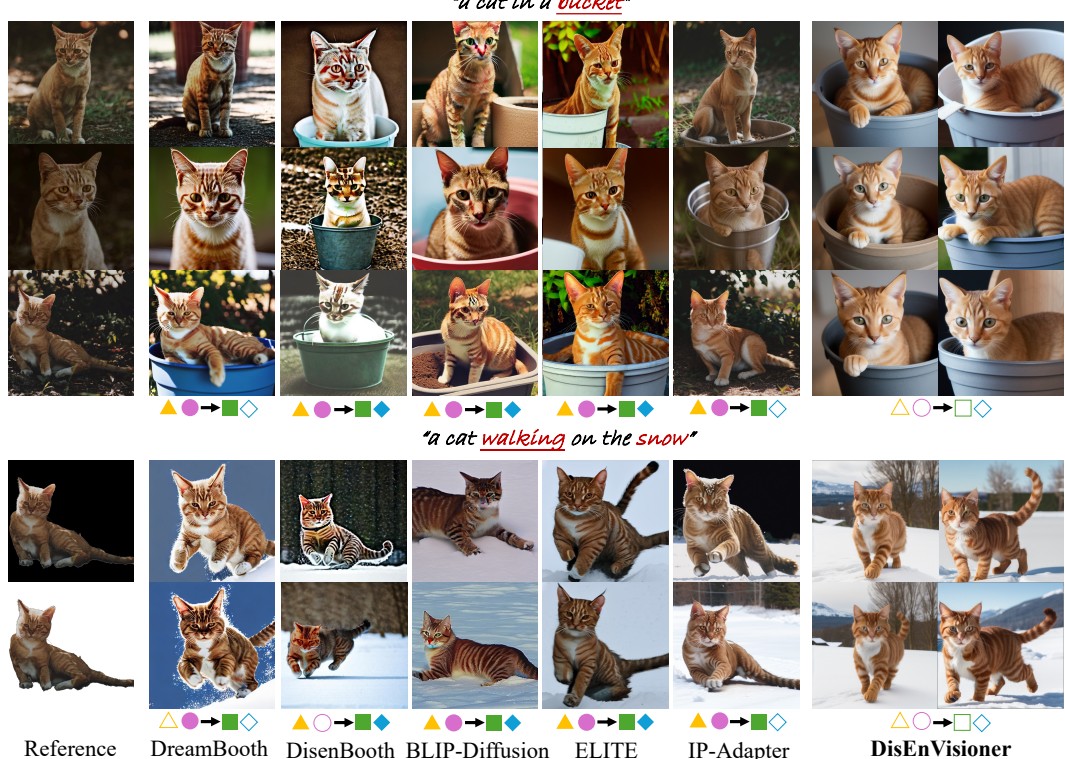

Figure 2: **Comparisons between DisEnvisioner and existing methods** (Ruiz et al., 2023; Chen et al., 2023a; Li et al., 2024; Wei et al., 2023; Ye et al., 2023) **under single-image setting**. We evaluate these methods on the same subject with different poses and environments. It can be observed that irrelevant factors, such as the subject's posture (▲) and background (●), can affect the customization quality and result in poor editability (■) or poor ID consistency (◆). For instance, BLIP-Diffusion falls in both two factors, leading to poor editability and ID consistency. We denotes its performance as "▲●→■◆" (the symbols without color filling, such as "△", indicates that the customization is *not* affected by subject's posture, and "□" indicates *good* editability). We also try to use masks to filter out irrelevant information for these methods. However, the harmful influence of subject's posture still exists. And solid background colors (*e.g.*, white or black) also can harmfully impact the customization quality, leading to textureless backgrounds.

# 1 INTRODUCTION

By training with billions of image-text pairs, state-of-the-art text-to-image generation models, *e.g.*, DALL·E (Ramesh et al., 2021), Imagen (Saharia et al., 2022), UnCLIP (Ramesh et al., 2022), Stable Diffusion (SD) (Rombach et al., 2022), and PixArt-$\alpha$ (Chen et al., 2023b), have demonstrated remarkable proficiency in generating contextually aligned images from textual descriptions. Despite unprecedented creative capabilities of these text-to-image models, customized image generation poses a new and more intricate challenge. This task aims to synthesize life-like imagery that not only accurately responds to natural language instructions (**editability**) but also preserves the subject's identity based on reference images (**ID consistency**), is garnering great attention from both academia and industry (Chen et al., 2023a; Dong et al., 2022; Shi et al., 2023; Ma et al., 2023; Gal et al., 2022; Kumari et al., 2023; Ruiz et al., 2023; Li et al., 2024; Wei et al., 2023; Ye et al., 2023; Li et al., 2023c; Wang et al., 2024; Chen et al., 2023c; Arar et al., 2024; Voynov et al., 2023).

For high-quality image customization, accurate interpretation of the visual prompt (*i.e.*, the input image) is crucial. This involves effectively extracting **subject-essential** attributes from the reference image while minimizing the influence of **subject-irrelevant** attributes. Failure to do so may result in 1) overemphasis on irrelevant details: generated images may prioritize irrelevant information, sidelining the textual instructions and compromising the overall editability; 2) diminished subject identity: the feature of subject-essential attributes becomes entangled with irrelevant information, degrading the subject representation.

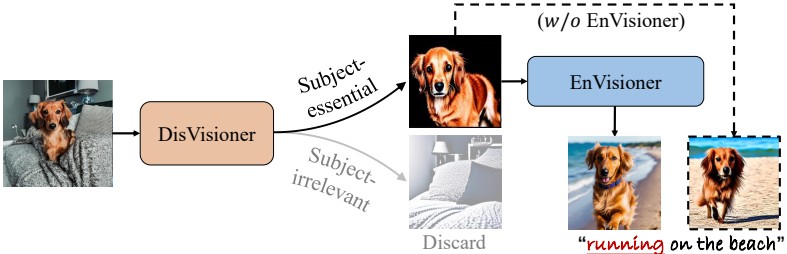

Figure 3: **Overview of DisEnvisioner**, which consists of two key components: **DisVisioner** and **En-Visioner**. During inference, subject-irrelevant features are discarded to avoid harmful disturbances.

Existing methods, both tuning-based (Ruiz et al., 2023; Gal et al., 2022; Kumari et al., 2023; Chen et al., 2023a; Dong et al., 2022) and tuning-free (Wei et al., 2023; Li et al., 2024; Ye et al., 2023; Ma et al., 2023; Li et al., 2023c), struggle to accurately interpret subject-essential attributes, particularly given only a single reference image. Specifically, prevailing tuning-based methods like DreamBooth (Ruiz et al., 2023) and DisenBooth (Chen et al., 2023a), heavily rely on multiple reference images to capture the common subject concept into the model. Despite their impressive results, it is difficult to interpret the subject-essential attributes in single-image scenarios, leading to compromised customization quality (Fig. 2). Additionally, each subject requires individual fine-tuning, which is time-consuming and hinders practical application. Recent tuning-free methods (Wei et al., 2023; Ma et al., 2023; Li et al., 2023c; Shi et al., 2023; Ye et al., 2023; Li et al., 2024; Chen et al., 2023c), aim to offer a significant boost in inference speed, and can generate customized images with a single reference image. However, these approaches still fail to accurately disentangle the subject-essential features from the reference image. Among those methods, IP-adapter (Ye et al., 2023) and PhotoVerse (Chen et al., 2023c) directly treat the *global* feature of the given image as *subject-essential*, inevitably introducing the subject-irrelevant information that diminishes the personalizing quality. Moreover, ELITE (Wei et al., 2023) and BLIP-Diffusion (Li et al., 2024) attempt to learn subject representations, but their implicit extraction of subject features proves ineffective (please see Sec. 2.2 for detailed discussion), also leading to unsatisfactory customization performance hindered by irrelevant information. As illustrated in Fig. 2, the customization quality of these methods are notably compromised by irrelevant factors (we conclude the primary irrelevant factors in this case as: subject's posture, background and image tone). A direct solution of these methods to prevent the irrelevant information is to use a segmentation mask to remove the background, but it is inadequate. As shown in Fig. 2: ① factors like the subject's pose still introduce irrelevant details; ② replacing the background with solid colors (*e.g.*, white or black) also influence the customization, potential leading to texture-less backgrounds in the generated customized images. Thus, an effective disentanglement of the subject-essential features is indeed necessary.

Motivated by the above analysis, we propose **DisEnvisioner**, a novel framework meticulously designed to addresses the core issues via feature disentanglement and enrichment. As illustrated in Fig. 3, the image is tokenized into compact disentangled features, *i.e.*, subject-essential and subject-irrelevant tokens, through the DisVisioner. Thus, the subject-irrelevant features can be filtered, making sure the model only focus on essential attributes of the subject, facilitating more accurate editability during customization. The disentangled subject-essential features are further enriched by EnVisioner before feeding into the pre-trained SD model (Rombach et al., 2022) for customized generation, significantly boosting the ID-consistency and the overall customization quality. Together with above innovations, we achieve both accurate and high-quality image customization.

In summary, our key contributions are as follows.

- We emphasize the critical role of subject-essential attribute in customized image generation, which is the foundation of faithful subject concept reconstruction and reliable editability, thereby ensuring more accurate customization under more diverse textual instructions.

- We present DisEnvisioner, a simple yet effective framework designed for single-image, tuning-free image customization, utilizing visual disentanglement and enrichment.

- Comprehensive experiments validate DisEnvisioner's superiority in adhering to instructions, maintaining ID consistency, and inference speed, demonstrating its superior personalization capabilities and efficiency.

## 2 RELATED WORKS

### 2.1 TEXT-TO-IMAGE GENERATION

In the field of text-to-image generation, the evolution of methodologies has transitioned from Generative Adversarial Networks (GANs) (Goodfellow et al., 2014; Zhang et al., 2017; 2018; 2021; He et al., 2022; Karras et al., 2019; 2020; 2021) to advanced Diffusion Models (Ho et al., 2020; Ramesh et al., 2022; Saharia et al., 2022; Ramesh et al., 2021; Nichol et al., 2021; Chen et al., 2023b; He et al., 2024; Rombach et al., 2022). Early GAN-based models like StackGAN (Zhang et al., 2017; 2018), AttnGAN (Xu et al., 2018), and XMC-GAN (Zhang et al., 2021) introduced multi-stage generation and attention mechanisms to improve image fidelity and alignment with textual descriptions. A significant breakthrough, DALL·E (Ramesh et al., 2021), utilizes a transformer and auto-regressive model to merge text and image data, trained on 250M pairs for high-quality, intuitive image synthesis. Following this, a series of diffusion-based methods such as GLIDE (Nichol et al., 2021), DALL·E2 (Ramesh et al., 2022), and Imagen (Saharia et al., 2022) have been introduced, offering enhanced image quality and textual coherence. The Latent Diffusion Model (LDM) (Rombach et al., 2022), trained on 5 billion pairs, further enhanced training efficiency without compromising performance, becoming a community standard. Despite remarkable strides in generating images from textual descriptions, current methodologies fall short in rendering customized visual concepts from reference images. In our paper, we dive into customized image generation, extending the capabilities of existing text-to-image techniques to craft personalized visual concepts.

### 2.2 CUSTOMIZED IMAGE GENERATION

Existing methods in the field of customized image generation primarily fall into two categories: tuning-based and tuning-free. Tuning-based methods (Gal et al., 2022; Ruiz et al., 2023; Dong et al., 2022; Han et al., 2023; Kumari et al., 2023; Chen et al., 2023a; Hua et al., 2023) involve fine-tuning a pre-trained generative model with several reference images of a specific concept during test-time. Despite the effectiveness of these methods, their high computational demand and scalability challenges limit the practicality. In contrast, tuning-free methods (Jia et al., 2023; Shi et al., 2023; Li et al., 2024; Ma et al., 2023; Ye et al., 2023; Wei et al., 2023; Chen et al., 2023c; Li et al., 2023c; Wang et al., 2024) employ advanced encoders to represent customized visual concepts, enabling image generation without the need of test-time fine-tuning and leading to significantly improved efficiency. However, InstantBooth (Shi et al., 2023) and PhotoMaker (Li et al., 2023c) still rely on multiple reference images to ensure the ID fidelity, and more recent works start to focus on singe-image scenarios. IP-Adapter (Ye et al., 2023) introduces a coupled cross-attention mechanism that handles visual and textual prompts separately, allowing for fine-grained visual concept generation using only a single reference image. PhotoVerse (Chen et al., 2023c) employs a similar dual-attention strategy, with an additional identity loss to further enhance ID consistency. However, both IP-Adapter and PhotoVerse directly treat the global feature of the given image as subject-essential, inevitably introducing the subject-irrelevant information that diminishes the personalizing quality. BLIP-Diffusion (Li et al., 2024) and ELITE (Wei et al., 2023) attempt to learn subject-essential representations. BLIP-Diffusion derives subject representations by querying image features with subject names using BLIP-2 (Li et al., 2023a). However BLIP-2 is pre-trained to extract *global* image feature aligned with *global* text prompts. It may struggle to accurately query clean *local* subject-essential features when the given subject names describe only the *local* information of a whole image. Thus, BLIP-Diffusion may still fall short in filtering out subject-irrelevant attributes. ELITE (Wei et al., 2023) separates subject attributes from others by utilizing CLIP features from different layers, assuming that the subject is represented by the deeper features and other concepts by shallower features. However, since the deep and shallow features are not fully disentangled, the subject attributes obtained through this multi-layer approach still remains inaccurate. Additionally, ELITE requires segmentation masks to combine the masked CLIP image features and the obtained subject feature for enhanced details. However, the segmenting process is cumbersome and the masked image feature is still inaccurate and can introduce subject-irrelevant factors. In this paper, although DisEnvisioner also adopts a two-stage pipeline following BLIP-Diffusion and ELITE, unlike these methods, DisEnvisioner focuses more on explicitly extracting the subject-essential attributes via DisVionser (Sec. 3.2), achieving more accurate customization. Then, we enrich the disentangled features by EnVisioner (Sec. 3.3), rather than introducing additional features as in ELITE (Wei et al., 2023), to further enhance the ID consistency without unnecessary disturbance.

## 3 METHOD

The goal of customized image generation is to synthesize lifelike images that adhere to the textual instructions while preserving the subject's identity from the reference image. To tackle the pivotal challenge of minimizing the influence of subject-irrelevant attributes and enhancing subject-essential attributes, we introduce DisEnvisioner—a novel approach focused on disentangling and enriching subject-essential attributes. As depicted in Fig. 3, DisEnvisioner initiates the process by disentangling the image features into subject-essential and -irrelevant attributes by the DisVisioner. Subsequently, to bolster the consistency of the subject's identity, the EnVisioner is further employed to refine the subject-essential features into a more fine-grained representation. Our discussion commences with an overview of basic concepts (Sec. 3.1), followed by an in-depth exploration of the DisVisioner (Sec. 3.2) and EnVisioner (Sec. 3.3).

### 3.1 PRELIMINARIES

Diffusion models (Ho et al., 2020; Ramesh et al., 2022; Saharia et al., 2022; Ramesh et al., 2021; Nichol et al., 2021; Chen et al., 2023b) represent the current state-of-the-art in generative modeling for high-fidelity image generation, it involves forward diffusion and reverse denoising processes. The forward process incrementally introduces noise to the data, transforming it into a Gaussian random noise by a Markov chain over a fixed number of steps $T$. In the reverse phase, a learned neural network is utilized to predict the added noise at each step, thereby recovering the original data from the noisy sample. This noising-denoising mechanism enables diffusion models to achieve impressive results in generating high-quality images with fine details and realism.

In this study, we adopt Stable Diffusion (SD) (Rombach et al., 2022), a latent diffusion model built upon UNet (Ronneberger et al., 2015), as our foundational generative model. Firstly, an auto-encoder (VAE) $\{\mathscr{E}(\cdot), \mathscr{D}(\cdot)\}$ is trained to map between RGB space and the latent space, $i.e.$, $\mathscr{E}(\mathbf{x}) = \mathbf{z}$, $\mathscr{D}(\mathscr{E}(\mathbf{x})) \approx \mathbf{x}$. And the textual conditions are obtained using a pre-trained CLIP text encoder $\boldsymbol{c} = \psi_\theta^{\mathrm{T}}(y)$, where $y$ is the given prompt. The training objective is:

$$L_{\mathrm{LDM}}(\theta) = \mathbb{E}_{\mathbf{x}_0, \boldsymbol{c}, \boldsymbol{\epsilon}_t \sim \mathcal{N}(0,1), t \in [1,T]} \left[ \|\boldsymbol{\epsilon}_t - \boldsymbol{\epsilon}_\theta(\mathbf{z}_t, \boldsymbol{c}, t)\|_2^2 \right], \tag{1}$$

where $\boldsymbol{\epsilon}_t$ represents the noise introduced during the forward process, and $\boldsymbol{\epsilon}_\theta(\mathbf{z}_t, \boldsymbol{c}, t)$ is the predicted noise. Cross-attention is adopted to introduce textual condition into SD. Specifically, the latent image feature $\boldsymbol{f} \in \mathbb{R}^{(\mathrm{H} \times \mathrm{W}) \times d_q}$ and text condition $\boldsymbol{c} \in \mathbb{R}^{n_y \times d_k}$ are firstly projected to obtain query $\boldsymbol{Q} = \boldsymbol{w}_{\mathrm{to\_q}} \circ \boldsymbol{f}$, key $\boldsymbol{K} = \boldsymbol{w}_{\mathrm{to\_k}} \circ \boldsymbol{c}$, and value $\boldsymbol{V} = \boldsymbol{w}_{\mathrm{to\_v}} \circ \boldsymbol{c}$, where $\boldsymbol{w}$ are weights of the corresponding mapping layers. Then, the cross-attention is calculated as:

$$\mathrm{Attention}(\boldsymbol{Q}, \boldsymbol{K}, \boldsymbol{V}) = \mathrm{SoftMax}\left(\frac{\boldsymbol{Q}\boldsymbol{K}^{\mathrm{T}}}{\sqrt{d_k}}\right)\boldsymbol{V}. \tag{2}$$

During inference, the model iteratively constructs images $\mathbf{x}_0$ from random noises $\mathbf{z}_T$ through reverse denoising process.

### 3.2 DISVISIONER

As revealed in (Wei et al., 2023; Li et al., 2024; Ye et al., 2023), the customized image can be encoded as a sequence of tokens via inversion. In our work, we propose to disentangle the image feature into subject-essential and -irrelevant tokens using an image tokenizer (Wu et al., 2022). Image tokenizer is a method to aggregate the image features into compact visual tokens, with each token corresponds to a distinct visual component. As illustrated in Fig. 4a, given a reference image $\mathbf{x}_{\mathrm{ref}}$, we firstly transform them using an augmentation set. The transformed image $\mathbf{x}'_{\mathrm{ref}}$ ensures that the model can extract the effective subject-essential feature to reconstruct the original image, rather than merely duplicating the subject. Subsequently, we employ the CLIP image encoder to extract image features $\boldsymbol{v}_{\mathrm{ref}} = \psi_\theta^{\mathrm{I}}(\mathbf{x}'_{\mathrm{ref}}) \in \mathbb{R}^{(\mathrm{H} \times \mathrm{W}) \times d_k}$. The image tokenizer network $M(\cdot)$ is then utilized to obtain the disentangled visual tokens $\mathcal{V}_{\mathrm{d}}$:

$$\mathcal{V}_{\mathrm{d}} = M(\boldsymbol{q}_{\mathrm{d}}, \psi_\theta^{\mathrm{I}}(\mathbf{x}'_{\mathrm{ref}})) \tag{3}$$

where $\boldsymbol{q}_{\mathrm{d}} \in \mathbb{R}^{(n_s + n_i) \times d_q}$ is the queries for feature aggregation, $n_s$ for the subject feature while $n_i$ for other irrelevant features. In image tokenizer $M(\cdot)$, a spatial-wise cross-attention mechanism is

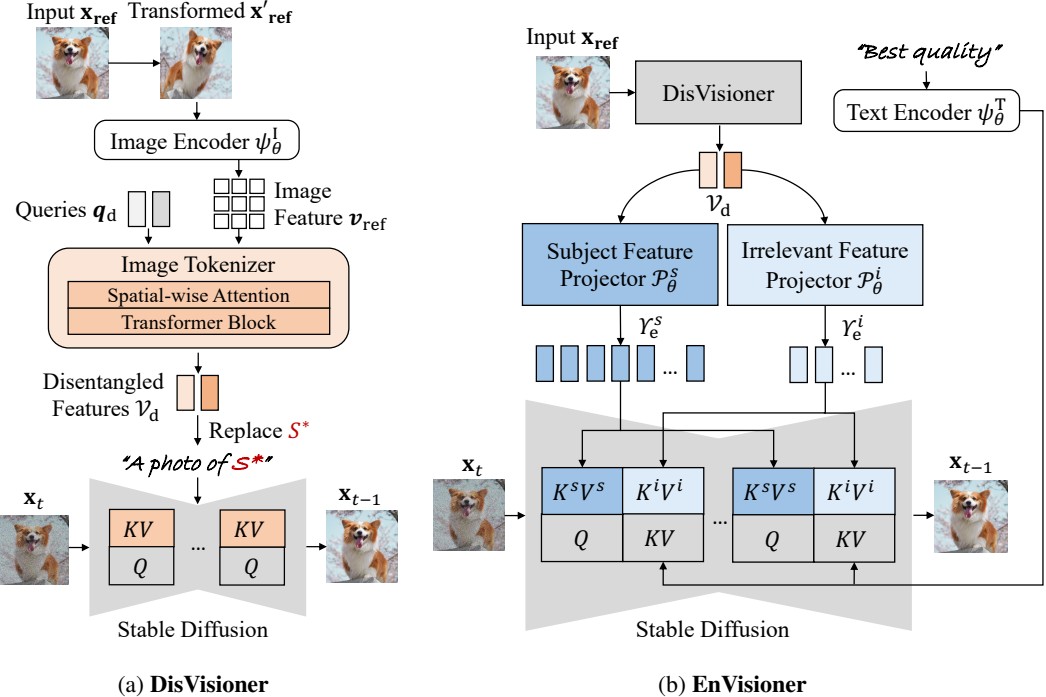

(a) **DisVisioner**          (b) **EnVisioner**

Figure 4: **Training pipeline of DisEnvisioner.** Our approach is structured into two stages: (a) Dis-Visioner firstly disentangles the features of the subject and other irrelevant components by aggregating the image feature $v_{\text{ref}}$ into two distinct and orthogonal tokens $v_{\text{d}}$. (b) EnVisioner subsequently refines and sculpts the disentangled features $\mathcal{V}_{\text{d}}$ into more granular representations to produce high ID-consistency images with the input image $\mathbf{x}_{\text{ref}}$, and can improve the overall visual quality of the image. Only colored modules (orange and blue) are trainable.

firstly adopted to aggregate the image features, where $q_{\text{d}}$ is the query and $\psi_\theta^{\text{I}}(\mathbf{x}_{\text{ref}})$ serves as the key and value. The image features are separated into two mutually independent and orthogonal sets of features according to $q_{\text{d}}$ after the spatial-wise attention mechanism. This mutual independence and orthogonality are ensured by the $\text{SoftMax}(\cdot)$ function applied at spatial dimension within the spatial-wise attention. In order to determine the sequence of disentangled features, we initialize $q_{\text{d}}$ with the CLIP prior, as revealed in (Li et al., 2023b). Specificallt, the $n_s$ queries for subject are initialized with random vectors, while the remaining $n_i$ queries are initialized using CLIP class name embeddings to query the subject-irrelevant tokens. Transformer blocks are followed to further refine the disentangled features to finally obtain $\mathcal{V}_{\text{d}}$ as the output of our DisVisioner.

To train the image tokenizer $M(\cdot)$, we insert the disentangled features $\mathcal{V}_{\text{d}}$ into textual feature space of the prompt by replacing the textual embedding of placeholder $S^*$, and adopt the Eq. 1 as the training objective for the target of reconstruction. Following (Wei et al., 2023; Kumari et al., 2023), the entire SD model is frozen except the `to_k` and `to_v` layers of the cross-attention module for correct interpretation of the new disentangled tokens. In our implementation, we set $n_s = 1$ and $n_i = 1$ for subject-essential and irrelevant features, respectively. Excessive tokens will lead to inaccurate disentanglement, thus impairing the customization quality (please refer to Sec. 4.4 for more details).

Benefiting from image tokenizer, the subject-essential features are accurately compressed into the tokens, clearly and accurately separated from irrelevant features via the spatial-wise attention space. During customized image generation, the subject-irrelevant token will be discarded for exclude the unwanted disturbance, facilitating the editing accuracy and ID fidelity.

## 3.3 ENVISIONER

While DisVisioner effectively extracts the subject-essential feature into a single token, it may be inadequate for capturing the detailed nuances of the customized subject. In EnVisioner, we map the disentangled features into a sequence of granular tokens using new projectors $P^s(\cdot)$ and $P^i(\cdot)$. The utilization of separate projectors also guarantees the disentanglement between the subject-essential and -irrelevant tokens.

As shown in Fig. 4b, the disentangled tokens $\mathcal{V}_{\mathrm{d}}$ from DisVisioner is enriched into multiple tokens:

$$\Upsilon_{\mathrm{e}}^{s} = P^{s} \circ \boldsymbol{\tau}_{\mathrm{d}}^{s}, \ \Upsilon_{\mathrm{e}}^{i} = P^{i} \circ \boldsymbol{\tau}_{\mathrm{d}}^{i}, \tag{4}$$

where $\tau_{\mathrm{d}}^{s}, \tau_{\mathrm{d}}^{i} \in \mathbb{R}^{1 \times d}$ denote the disentangled subject-essential and the irrelevant feature through DisVisioner ($\mathcal{V}_{\mathrm{d}} = [\tau_{\mathrm{d}}^{s}, \tau_{\mathrm{d}}^{i}]$). $\Upsilon_{\mathrm{e}}^{s} \in \mathbb{R}^{n'_{s} \times \mathrm{d}}$ is the enriched subject-essential tokens from $\tau_{\mathrm{d}}^{s}$, and $\Upsilon_{\mathrm{e}}^{i} \in \mathbb{R}^{n'_{i} \times \mathrm{d}}$ is the irrelevant tokens projected from $\tau_{\mathrm{d}}^{i}$. By enriching the subject-essential token into multiple ones, the disentangled concept is further enhanced into a more granular representation, improving especially the ID consistency between the synthesized image and the reference image.

Similar to DisVisioner, we also employ Eq. 1 as the training objective to train the feature projectors in EnVisioner, while keeping the entire SD model and all DisVisioner modules frozen. Additionally, we separatelt introduce cross-attention layers for subject-essential tokens and -irrelevant ones. This separate injection strategy further ensures that the subject feature will not be interfered with by other irrelevant factors. Specifically, given the latent diffusion feature $\boldsymbol{f} \in \mathbb{R}^{(\mathrm{H} \times \mathrm{W}) \times d_{\mathrm{q}}}$ and text instruction $\boldsymbol{c} \in \mathbb{R}^{n_{y} \times d_{\mathrm{k}}}$, the cross-attention output $\boldsymbol{f}'$ is derived through three decoupled cross-attention layers, which can be described via the following Equation:

$$\boldsymbol{f}' = \mathrm{Attention}(\boldsymbol{Q}, \boldsymbol{K}, \boldsymbol{V}) + \lambda_{s} \, \mathrm{Attention}(\boldsymbol{Q}, \boldsymbol{K}^{s}, \boldsymbol{V}^{s}) + \lambda_{i} \, \mathrm{Attention}(\boldsymbol{Q}, \boldsymbol{K}^{i}, \boldsymbol{V}^{i}),$$
$$\text{where } K^{s} = \boldsymbol{w}_{\mathrm{to\_k}}^{s} \circ \Upsilon_{\mathrm{e}}^{s}, V^{s} = \boldsymbol{w}_{\mathrm{to\_v}}^{s} \circ \Upsilon_{\mathrm{e}}^{s}; K^{i} = \boldsymbol{w}_{\mathrm{to\_k}}^{i} \circ \Upsilon_{\mathrm{e}}^{i}, V^{i} = \boldsymbol{w}_{\mathrm{to\_v}}^{i} \circ \Upsilon_{\mathrm{e}}^{i}, \tag{5}$$

the $\mathrm{Attention}(\cdot)$ is defined in Eq. 2, $\boldsymbol{w}_{\mathrm{to\_k}}^{s}, \boldsymbol{w}_{\mathrm{to\_v}}^{s}, \boldsymbol{w}_{\mathrm{to\_k}}^{i}$ and $\boldsymbol{w}_{\mathrm{to\_v}}^{i}$ are trainable mapping layers. During training, weights $\lambda_{s}$ and $\lambda_{i}$ are fixed at 1.0. During inference, for the purpose of effectively ignoring subject-irrelevant features encoded in $\Upsilon_{\mathrm{e}}^{i}$, we set $\lambda_{i} = 0$.

## 4 EXPERIMENTS

### 4.1 EXPERIMENTAL SETUP

#### 4.1.1 TRAINING DATASET

We use the *training set* of OpenImages V6 (Kuznetsova et al., 2020) to train the DisEnvisioner. It contains about 14.61M annotated boxes across 1.74M images. Based on this dataset, we construct 6.82M {prompt, image} pairs for training, where the images are cropped and resized ($256 \times 256$ for DisVisioner and $512 \times 512$ for EnVisioner) according to the bounding box annotations, and the text prompts are obtained by randomly selecting a CLIP ImageNet template (Radford et al., 2021) and integrating the annotated class names into it.

#### 4.1.2 EVALUATION DATASET AND METRICS

The evaluation is carried out on the DreamBooth (Ruiz et al., 2023) dataset, which comprises 30 subjects and 158 images in total (4∼6 images per subject). For quantitative evaluation, 25 editing prompts (Ruiz et al., 2023) are used for each image. We inference 40 times for each {prompt, image} pair, generating 158,000 customized images for evaluation across 6 metrics.

In alignment with previous methodologies (Ruiz et al., 2023; Kumari et al., 2023; Wei et al., 2023; Li et al., 2023c), we utilize: 1) CLIP Text-alignment (**C-T**) to measure the instruction response fidelity; 2) CLIP Image-alignment (**C-I**) and 3) DINO Image-alignment (**D-I**) (Caron et al., 2021) to evaluate the ID consistency with the reference image. Additionally, we introduce a novel metric, *i.e.*, Internal Variance (**IV**) of image-alignment, aiming at quantifying the impact of irrelevant factors. IV measures the variance of customization results for images containing the same subject but different environments (like the top three rows in Fig. 2). Lower IV value indicates that the results only be controlled by the subject and textual instruction, rather than the irrelevant factors from environments. In terms of efficiency, we record the 6) Inference-time (**T**) on single NVIDIA A800 GPU for evaluation. Moreover, we calculate the mean ranking (**mRank**) of all metrics for each method to show the comprehensive performance.

#### 4.1.3 IMPLEMENTATION DETAILS.

DisEnvisioner is built upon Stable Diffusion v1.5 , employing OpenCLIP ViT-H/14 model as the image/text encoder. During training, DisVisioner is configured with batch size of 160, learning rate of 5e-7 at the resolution of 256. The training steps is 120K. We set the token number $n_{s} = 1$

Table 1: **Quantitative comparisons with existing methods**. The evaluation metrics include text alignment for assessing editability (C-T), image-alignment for ID-consistency (C-I, D-I), internal variance to demonstrate the resistance to subject-irrelevant factors (IV), and inference time for efficiency (T). DisEnvisioner demonstrates better comprehensive performance than other methods. Top results are in **bold**; second-best are underlined.*For equity, we assess only inference (and test-time tuning, if applicable) times, omitting I/O operations. §For equity, we consider C-I and D-I as two sub-indicators of image-alignment, the rank of each with a weight of **0.5** in the mRank calculation, while the ranks of all other metrics have a weight of **1.0**.

| Method | C-T↑ | C-I§↑ | D-I§↑ | IV↓ | T*↓ (s) | mRank↓ |
|---|---|---|---|---|---|---|
| DisenBooth (Chen et al., 2023a) | 0.303 | 0.760 | 0.781 | 0.041 | 2.42e+3 | 4.8 |
| DreamBooth (Ruiz et al., 2023) | 0.286 | 0.842 | 0.849 | 0.039 | 1.12e+3 | 4.3 |
| ELITE (Wei et al., 2023) | 0.287 | 0.792 | 0.770 | 0.036 | 4.12 | 4.1 |
| IP-Adapter (Ye et al., 2023) | 0.275 | **0.883** | **0.912** | 0.033 | 1.98 | 3.3 |
| BLIP-Diffusion (Li et al., 2024) | 0.295 | 0.785 | 0.765 | 0.029 | **1.10** | 2.9 |
| **DisEnvisioner** | **0.315** | 0.828 | 0.802 | **0.026** | 1.96 | **2.0** |

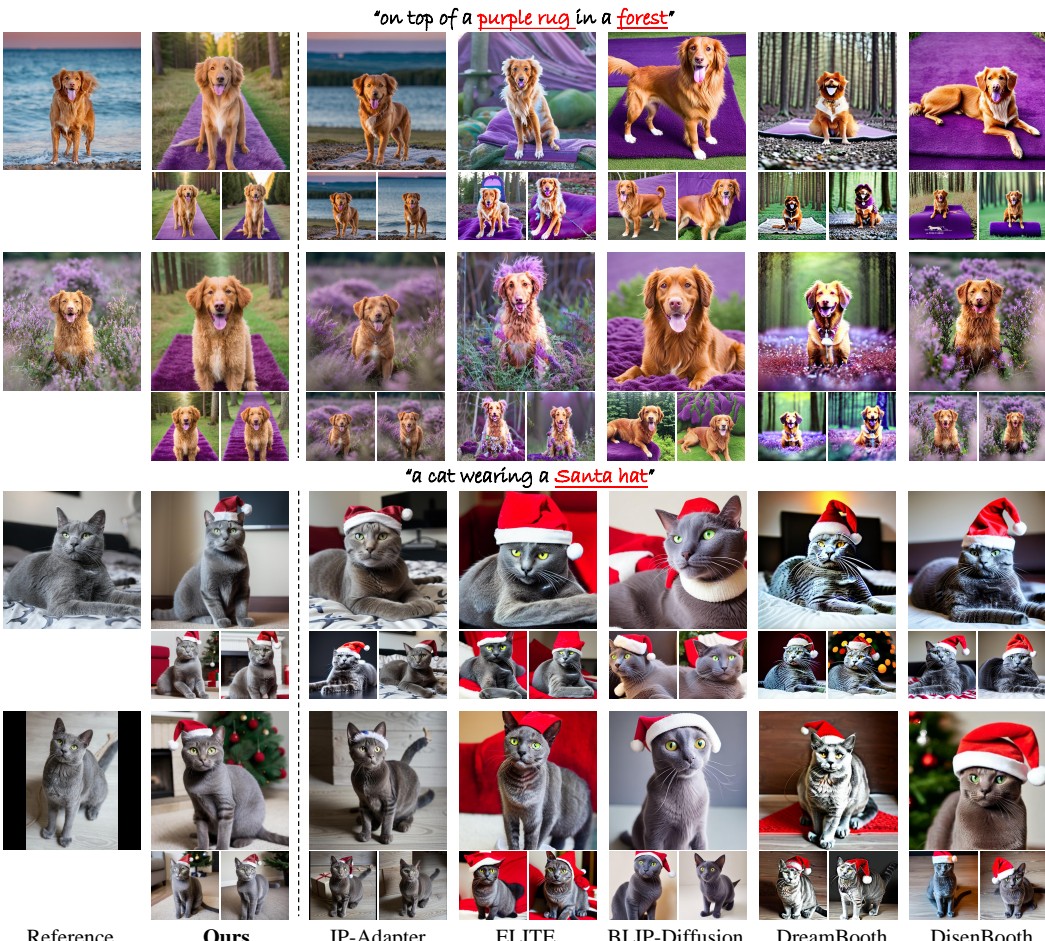

Figure 5: **Qualitative comparison on live subjects.** Comparing two subjects across various scenarios, DisEnvisioner excels in editability and ID-consistency. Notably, the animal's posture does not affect customization, showcasing our strength in capturing subject-essential features.

and $n_i = 1$, for subject-essential feature and -irrelevant respectively. The EnVisioner employs the batch size of 40, learning rate of 1e-4 at the resolution of 512. The training steps is also 120K. The enriched token number is $n'_s = 4$ and $n'_i = 4$, with attention scale $\lambda_s = 1.0$ and $\lambda_i = 1.0$. All experiments are conducted on 8 NVIDIA A800 GPUs using the AdamW optimizer (Loshchilov & Hutter, 2017) with a weight decay of 0.01. To enable classifier-free guidance (Dhariwal & Nichol, 2021), we use a probability of 0.05 to drop the condition, both textual and visual. During inference, $\lambda_i$ is set to 0 to eliminate irrelevant feature. In addition, we use the DDIM sampler (Song et al., 2020) with 50 steps and the scale of classifier-free guidance is set to 5.0.

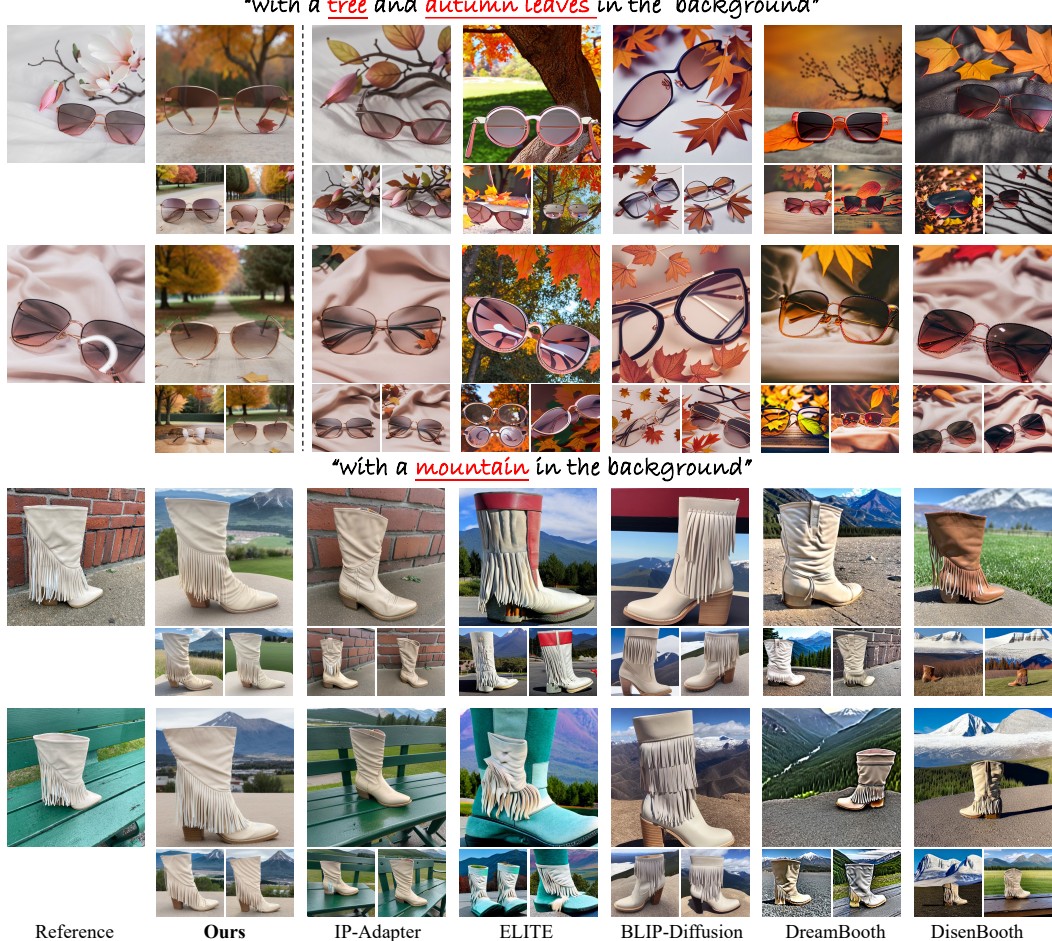

Figure 6: **Qualitative comparison on non-live subjects.** Besides offering better editability of textual instructions, DisEnvisioner also excels in preserving accurate subject identity. Moreover, images generated by DisEnvisioner are minimally affected by irrelevant elements of reference images.

## 4.2 QUANTITATIVE RESULTS

We compare DisEnvisioner against five leading methods according to metrics specified in Sec. 4.1.2. For the tuning-based methods, we chose DreamBooth (Ruiz et al., 2023) and DisenBooth (Chen et al., 2023a), and implement them within the single-image setting. Our tuning-free comparison covers all available open-source methods, they are ELITE (Wei et al., 2023), BLIP-Diffusion (Li et al., 2024), and IP-Adapter (Ye et al., 2023). As demonstrated in Tab. 1, our approach achieves the highest text-alignment score (C-T), indicating its effectiveness in eliminating subject-irrelevant information and significantly enhancing editability. Although IP-Adapter (Ye et al., 2023) and Dream-Booth (Ruiz et al., 2023) achieve high image-alignment scores (C-I and D-I), they struggle with editability. This is because they tend to replicate large portions of the reference image, resulting in excessive consistency and reduced flexibility in making edits (Fig. 5 and 6). Excluding these two methods, our approach achieves the highest image-alignment scores (C-I and D-I), demonstrating its superior ID consistency without compromising text-alignment. We also achieve lower internal variance (IV) between the generated images created with the same subject under different conditions, further demonstrating that DisEnvisioner is unaffected by irrelevant factors, such as subject pose and its surroundings. In terms of efficiency, thanks to our lightweight and effective design, eliminating the test-time tuning, only 1.96s is required for each customized generation. We also conduct a user study, as detailed in the supplementary materials, which further demonstrates the superiority of DisEnvisioner.

## 4.3 QUALITATIVE RESULTS

To obtain deeper insights of the DisEnvisioner, we visualize its synthesized images against the selected five prevailing methods across three different images per subject. Fig. 5 and 6 clearly

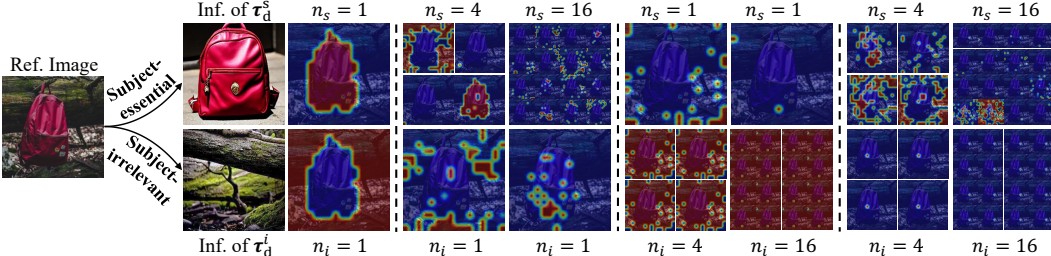

Figure 7: **Ablation on token's numbers in DisVisioner.** Each attention map is calculated by the dot product of the obtained token ($\tau_d^s, \tau_d^i$) and the CLIP local image features. The results demonstrates that $n_s = 1$ and $n_i = 1$ outperforms other configurations, achieving precise attention map and faithful token inference.

demonstrates DisEnvisioner's superiority in producing high-quality, editable images with strong ID-consistency. Notably, the consistency in animal postures alongside the minimal impact of irrelevant backgrounds across reference images under the same textual instruction, also showcase our robustness and resistance to subject-irrelevant elements, *i.e.*, only the subject-essential attributes are extracted and preserved. This excellence also applies to non-live objects, as seen in Fig. 6, where exceptional customizations for subject the boot and sunglasses demonstrate DisEnvisioner's accurate focus on subject-essential features, leading to superior editability, ID-consistency, and the overall visual quality of the generated images.

## 4.4 ABLATION STUDY

### 4.4.1 INFLUENCE OF TOKENS NUMBERS IN DISVISIONER.

As described in Sec. 3.2, the disentangled tokens are injected into the textual space of CLIP encoder, thus each token tends to represent a complete semantic meaning, similar to "words". Meanwhile, due to the mutual exclusivity between subject-essential and -irrelevant features in DisVisioner, an excessive number of tokens leads to increased competition, making disentanglement difficult and chaotic. For instance, in Wu et al. (2022), only four tokens are used to represent a complex outdoor scene, suggesting that fewer tokens may be more effective for simpler subjects. To achieve semantic completeness and avoid excessive feature competition, we conduct experiments on the number of subject-essential and -irrelevant tokens, *i.e.*, $n_s$ and $n_i$. As visualized in Fig. 7, when $n_s \neq n_i$, the

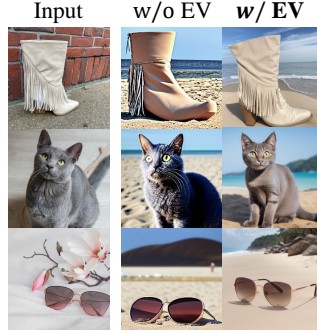

Figure 8: **Ablation of EV.**

image is represented by the larger set without any disentanglement. When $n_s = n_i > 1$, random initialized subject tokens have more "freedom", capturing most image information and also leading to reconstruction. We also observe that all $n_i$ tokens are the same, that is because they use the same prior discussed in Sec. 3.2. In summary, $n_i = 1$ and $n_s = 1$ performs the best.

### 4.4.2 EFFECT OF ENVISIONER.

As shown in Fig. 8, we validate the effect of enriched subject representation in EnVisioner (abbreviated as EV). The prompt is "*a photo of S\* on the beach*". We can observe that the images enhanced by EnVisioner exhibit finer details and improved ID-consistency, with a significant boost in overall image quality, emphasizing its crucial role in delivering high-quality customization.

## 5 CONCLUSION

In this paper, we propose **DisEnvisioner**, which is characterized by its emphasis on the interpretation of subject-essential attributes for high-quality image customization. DisEnvisioner effectively identifies and enhances the subject-essential feature while filtering out other irrelevant information, enabling exceptional image customization without cumbersome tuning or relying on multiple reference images. Through both the quantitative and qualitative evaluations, alongside the user study, we demonstrate DisEnvisioner' superior performance in customization quality and efficient inference time, offering a promising solution for practical applications.

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

## A  APPENDIX

You may include other additional sections here.

