# Supplementary Materials of
# DisEnvisioner: Disentangled and Enriched Visual Prompt for Customized Image Generation

**Jing He[1]\* Haodong Li[1]\* Yongzhe Hu Guibao Shen[1] Yingjie Cai[3] Weichao Qiu[3] Yingcong Chen[1,2] ✉**
[1]HKUST(GZ) [2]HKUST [3]Noah's Ark Lab
{jhe812, hli736}@connect.hkust-gz.edu.cn; yingcongchen@ust.hk

## A  User Study

Table A: **User study.** Participants rank the methods based on four criteria using a round-robin format, with the final scores are normalized before being reported. **mRank** is also reported.

| Method | TA↑ | IA↑ | IV↑ | IQ ↑ | mRank↓ |
|---|---|---|---|---|---|
| DreamBooth (Ruiz et al., 2023) | 0.159 | 0.152 | 0.159 | 0.158 | 3.5 |
| DisenBooth (Chen et al., 2023) | 0.157 | 0.143 | 0.156 | 0.140 | 5.3 |
| ELITE (Wei et al., 2023) | 0.161 | 0.147 | 0.154 | 0.147 | 4.3 |
| IP-Adapter (Ye et al., 2023) | 0.153 | 0.177 | 0.152 | 0.181 | 4.0 |
| BLIP-Diffusion (Li et al., 2024) | 0.159 | 0.162 | 0.163 | 0.165 | 2.8 |
| **DisEnvisioner** | **0.211** | **0.219** | **0.216** | **0.209** | **1.0** |

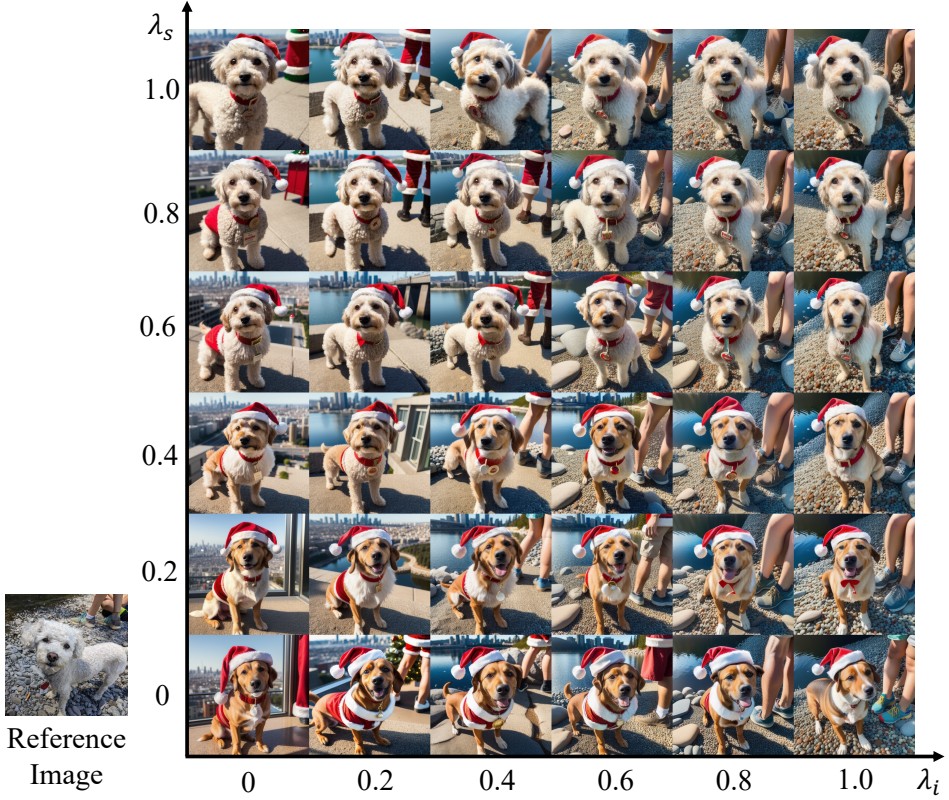

Figure A: **Effect of varying $\lambda_s$ and $\lambda_i$** with class name provided. The prompt is "a dog is wearing a Santa hat with a city in the background".

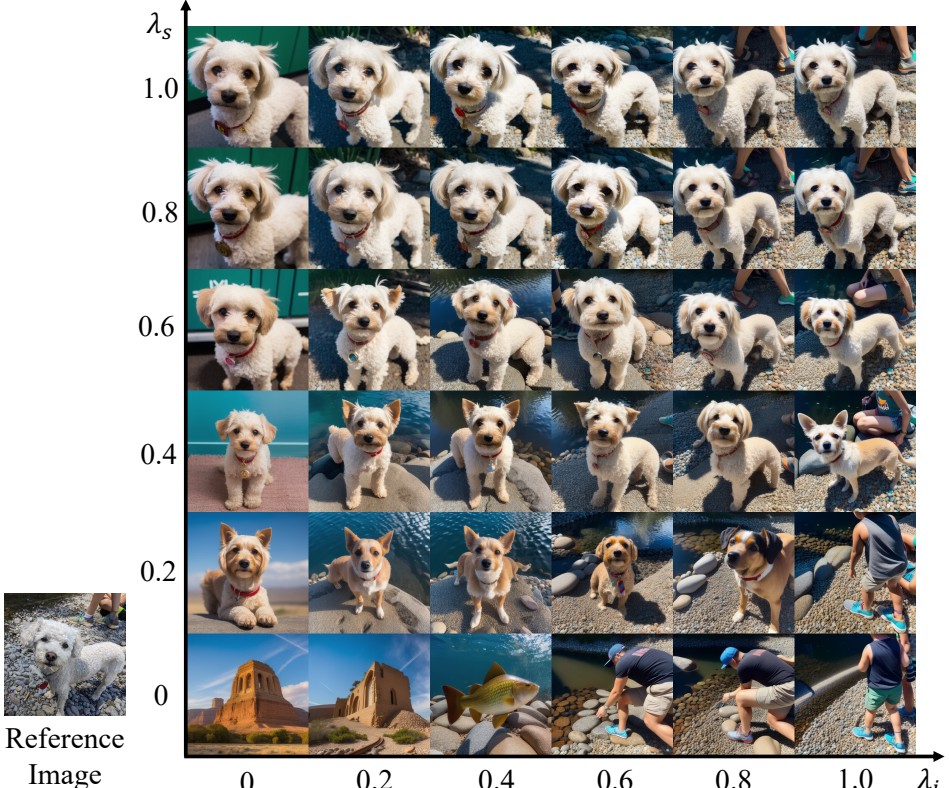

Figure B: **Effect of varying** $\lambda_s$ **and** $\lambda_i$ with *empty* prompts without providing class names.

We also conduct a user study using a round-robin format, where participants grade each method given the generation outputs across 3 images of the same subject in each round. There are 5 rounds in total. The metrics are text alignment (**TA**), subject identity alignment (**IA**), internal variance of those 3 customized images (**IV**), and image quality (**IQ**). A total of 69 users, and 345 rounds are recorded. As detailed in Table A, DisEnvisioner outperforms all baselines in all metrics.

During each round of the user study, rather than **ranking** DisEnvisioner and five other existing methods from the best to the worst, users are expected to **assign grades** from 0 to 5 to each method according to specific metrics. In practice, all participants have the complete freedom to grade any method with any score based on their personal judgment. After a total of 345 rounds of evaluation, the best-performing method often receives the highest scores, while scores for other methods are frequently identical due to similar customization quality. Additionally, it is uncommon for users to assign scores as low as 0 or 1. Although the grading differences among methods are not particularly large, DisEnvisioner consistently outperforms others competitors across all evaluation criteria.

## B  EFFECT OF $\lambda_s$ AND $\lambda_i$

As defined in Eq. 5 of the main paper, the weights $\lambda_s$ and $\lambda_i$ serve to modulate the integration of information that is essential and irrelevant to the subject in the given reference image. To thoroughly assess their effect of disentanglement, we adjust their values evenly sampled from 0 to 1.0 throughout the image customization process.

We generate images under varying settings of $\lambda_s$ and $\lambda_i$ employing both empty and non-empty (editing) prompts. Fig. B and C demonstrate that as $\lambda_i$ decreases progressively (moving from the right to the left columns), the presence of subject-irrelevant disturbances in the images notably declines. Also, enhancing $\lambda_s$ (moving from the bottom to the top rows) brings more pronounced consistency in subject identity between the reference and generated image. When both $\lambda_s$ and $\lambda_i$ are reduced to their minimum value, *i.e.*, $\lambda_s = 0$ and $\lambda_i = 0$, the generated images are solely influenced by the textual prompt, without incorporating any information from the reference image. To further explore the role of additional information in the prompt, we also generate images with specific class names included in the prompts.

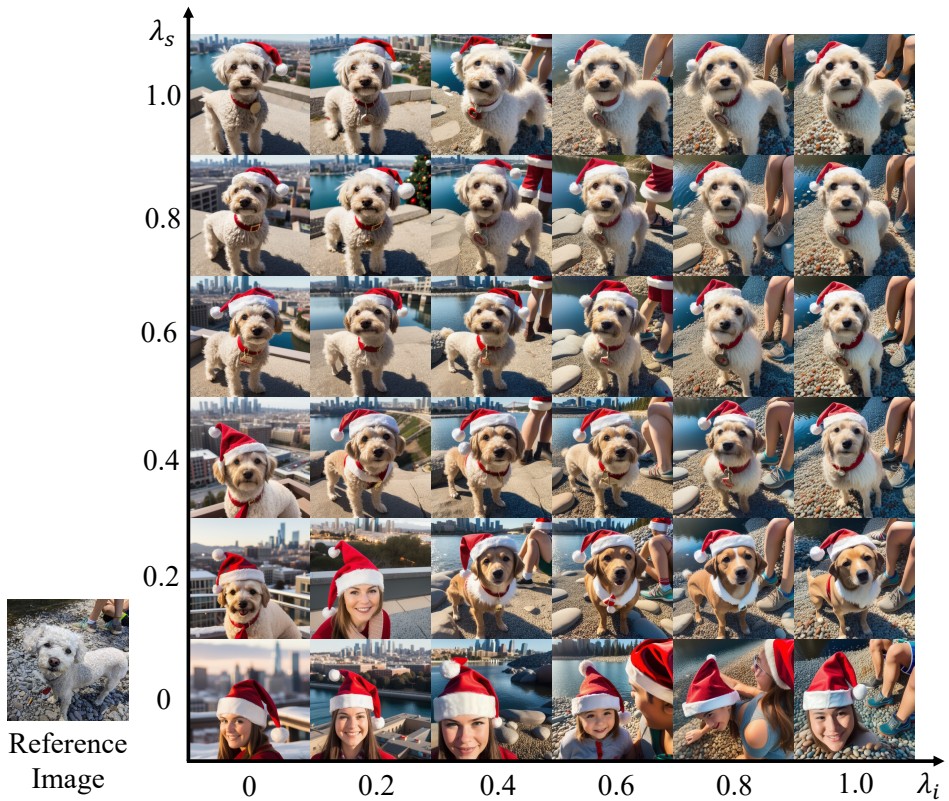

Figure C: **Effect of varying** $\lambda_s$ **and** $\lambda_i$ with *non-empty* prompts without providing class names.

As illustrated in Fig. A and Fig. C, particularly in terms of identity consistency, no matter the category-guidance (for instance, the class name "dog") is provided or not, it does not alter the customization quality. This indicates that DisEnvisioner effectively deciphers and extracts subject-essential attributes from the reference image. By doing so, it can accurately identify relevant and redundant information, which is then eliminated.

It is also evident that when image generation focuses exclusively on solely subject-essential features ($\lambda_s = 1.0$ and $\lambda_i = 0$) or solely on purely subject-irrelevant features ($\lambda_s = 0$ and $\lambda_i = 1.0$), the reproduction of the subject and the irrelevant surrounding content is achieved independently, devoid of any interference from one another. This phenomenon confirms the proficiency of DisEnvisioner in precisely segregating and enriching subject-essential features. It highlights the DisEnvisioner's exceptional customization performance without the need for test-time tuning, and relying solely on a single reference image.

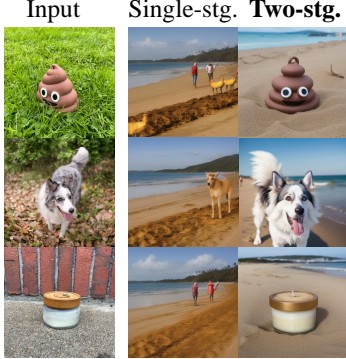

Figure G: **Comparison with single-stage and current two-stage training.** "stg." represents the training stage.

## C   CAN DISENVISIONER BE TRAINED IN SINGLE-STAGE?

For single-stage training, we combine the training processes of DisVisioner and EnVisioner, aiming for simultaneous learning the disentanglement and enrichment of subject-essential features. However, in Fig. G, the single-stage model is difficult to be trained. It fails to capture any subject-essential information from the visual prompt and merely response to text instructions. Nonetheless, the two-stage strategy of DisEnvisioner—separating disentanglement and enrichment—proves to be much more effective in high-quality customization.

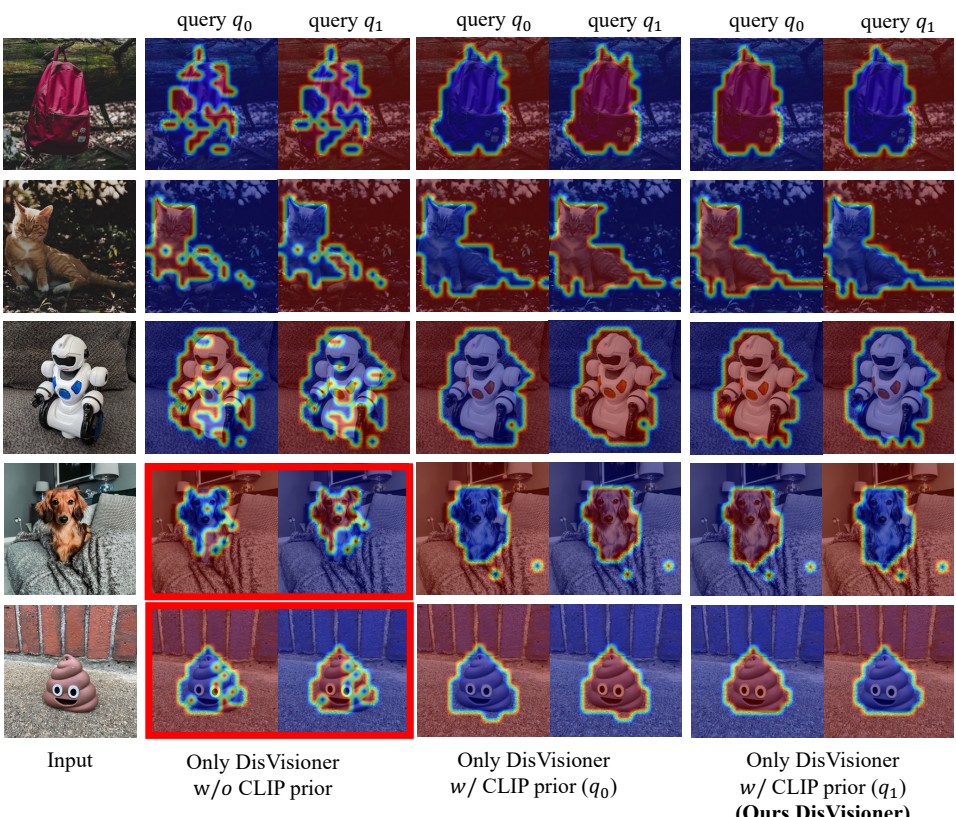

Figure D: **Ablation on CLIP prior initialization.** Each attention map is calculated by the dot product of the obtained token $(\tau_{\mathrm{d}}^s, \tau_{\mathrm{d}}^i)$ and the CLIP *local* image embeddings. The results demonstrate the effectiveness of the CLIP prior in identifying the subject token. The red borders highlight the uncertainty in the ordering of subject and irrelevant tokens.

Table B: **Ablation on CLIP prior initialization.** Best results are **bolded** and the second best are underlined. Please see main paper's Sec. 4.1.2 and Tab. 1 for specific definitions of those metrics. For equity, we consider C-I and D-I as two sub-indicators of image-alignment, the rank of each with a weight of **0.5** in the mRank calculation, while the ranks of all other metrics have a weight of **1.0**.

| Method | C-T↑ | C-I↑ | D-I↑ | I-V↓ | mRank↓ |
|---|---|---|---|---|---|
| DisVisioner - $w/o$ clip prior | 0.289 | 0.765 | 0.748 | 0.029 | 3.0 |
| DisVisioner - $w/$ clip prior $(q_0)$ | **0.314** | 0.786 | **0.770** | **0.026** | **1.2** |
| DisVisioner - $w/$ clip prior $(q_1)$ | **0.314** | **0.789** | 0.768 | **0.026** | **1.2** |

# D    ABLATION ON CLIP PRIOR INITIALIZATION

We conduct experiments to validate the effect of CLIP prior initialization. The "DisVisioner $w/o$ CLIP prior" was implemented by randomly initializing both subject and irrelevant tokens. As shown in Fig. D, although rough disentanglement is achieved, it becomes difficult to clearly identify the subject token. The red borders highlight the uncertainty in the ordering of subject and irrelevant tokens. We further initialize the tokens in different orders: "CLIP prior $(q_0)$" represents the first token is initialized with CLIP prior and "CLIP prior $(q_1)$" denotes the second token is initialized with CLIP prior. As shown in Fig. D, the CLIP prior effectively determines which token represents the subject-irrelevant information, and the effectiveness of disentanglement is further improved.

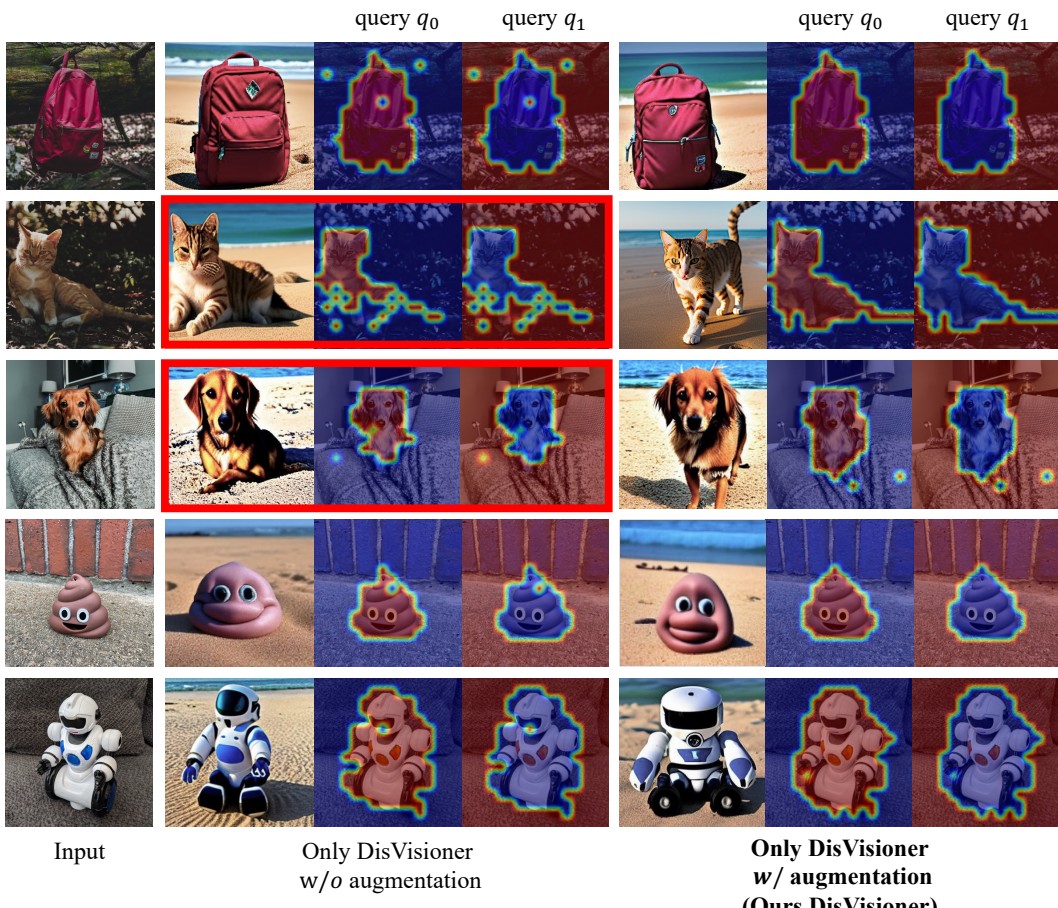

Figure E: **Ablation on the effect of augmentations in DisVisioner.** Each attention map is calculated by the dot product of the obtained token $(\tau_d^s, \tau_d^i)$ and the CLIP *local* image features. The red borders highlight the model tends to directly copy irrelevant patterns and semantics from the input image without employing augmentations, *e.g.*, the poses. All animal samples are generated with the prompt of "a * running on the beach", and object samples with "a * on the beach".

Table C: **Ablation on the effect of augmentations in DisVisioner**. The editability is poor without augmentations. Best results are **bolded**.

| Method | C-T↑ | C-I↑ | D-I↑ | I-V↓ | mRank↓ |
|---|---|---|---|---|---|
| DisVisioner, $w/o$ augmentation | 0.292 | 0.787 | 0.764 | 0.029 | 2.0 |
| DisVisioner, $w/$ augmentation | **0.314** | **0.789** | **0.768** | **0.026** | **1.0** |

In terms of quantitative comparisons, as shown in Tab. B, the performance of "CLIP prior $(q_0)$" and "CLIP prior $(q_1)$" (Ours DisVisioner) are very similar, as both effectively disentangle the subject and irrelevant tokens. When using "$w/o$ CLIP prior", since the subject token cannot be clearly identified, by default, we select the first token as the subject token. As illustrated in Tab. B, using "$w/o$ CLIP prior" results in degraded edibility and identity metrics. This is caused by the reduced quality of the subject-essential token, which sometimes represents irrelevant components.

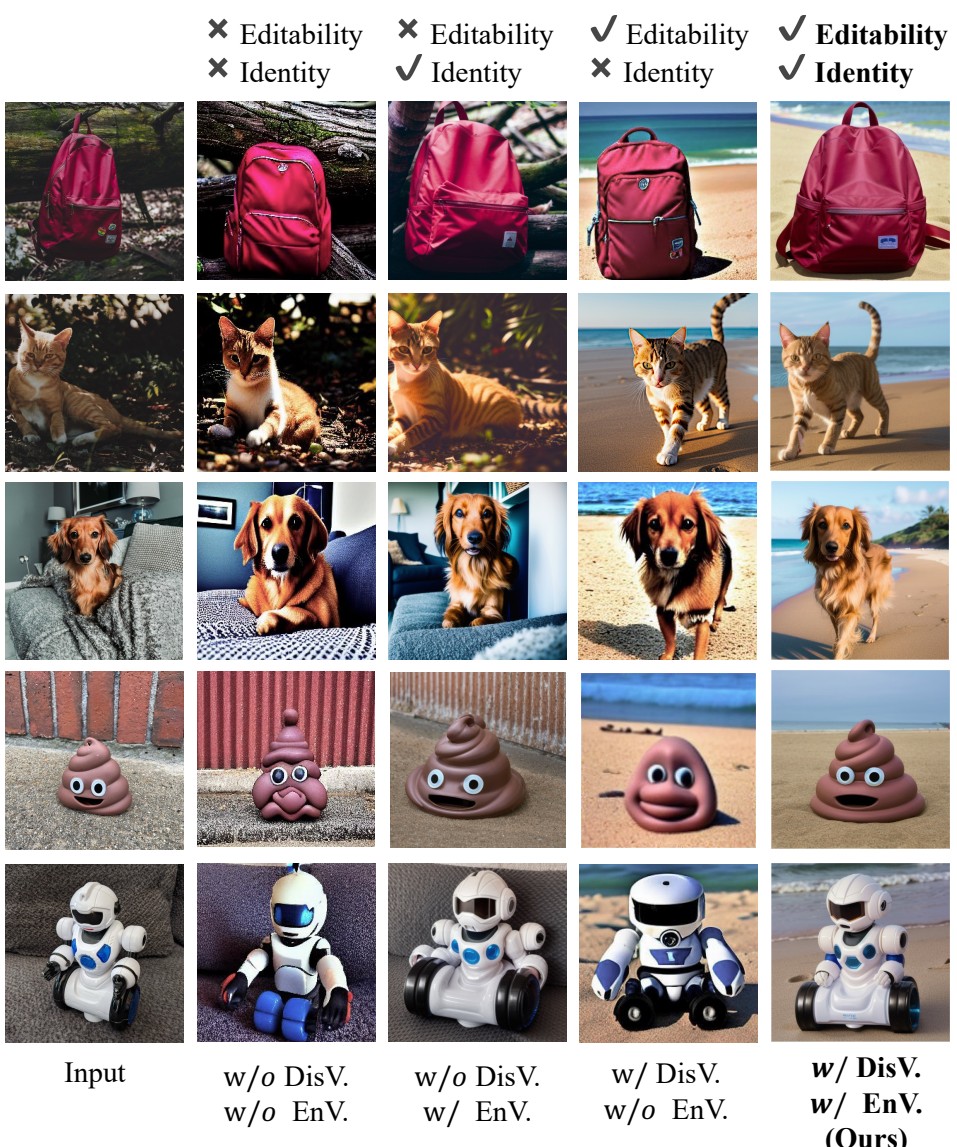

Figure F: **Ablation on DisVisioner and EnVisioner.** All animal samples are generated with the prompt of "a * running on the beach", and object samples with "a * on the beach". "DisV." represents the proposed DisVisioner, and "EnV." indicates the proposed EnVisioner.

## E    ABLATION ON THE EFFECT OF AUGMENTATIONS IN DISVISIONER

We conduct an ablation study by removing augmentations during the training of DisVisioner. As shown in Fig. E, although disentanglement is achieved through the tokenizer and CLIP prior, the generated images exhibit relative poor editability, as the model tends to directly copy irrelevant patterns and semantics from the input image, such as the poses of animals, the position of the subject in the reference image, and environmental lighting conditions. The red borders highlight these issues. When augmentations are applied during training, the model learns to extract essential subject features to reconstruct the original image under different transformations, rather than merely reconstructing exactly the same image.

Due to poor editability, "DisVisioner $w/o$ augmentation" results in lower C-T score, as shown in Tab. C. Additionally, since the model reproduces both subject-essential features and some irrelevant

Table D: **Ablation on DisVisioner and EnVisioner.** Both DisVisioner and EnVisioner yields effective performance improvements. Best results are **bolded** and the second best are underlined. For equity, we consider C-I and D-I as two sub-indicators of image-alignment, the rank of each with a weight of **0.5** in the mRank calculation, while the ranks of all other metrics have a weight of **1.0**. It is worthwhile to mention that high C-I and D-I scores are not always a good phenomenon when the editability (C-T) is poor.

| Method | C-T↑ | C-I↑ | D-I↑ | I-V↓ | mRank↓ |
|---|---|---|---|---|---|
| w/o DisVisioner, w/o EnVisioner | 0.260 | 0.796 | 0.771 | 0.041 | 3.3 |
| w/o DisVisioner, w/ EnVisioner | 0.262 | **0.871** | **0.805** | 0.042 | 2.3 |
| w/ DisVisioner, w/o EnVisioner | 0.314 | 0.789 | 0.768 | **0.026** | 2.3 |
| **w/ DisVisioner, w/ EnVisioner (Ours)** | **0.315** | 0.828 | 0.802 | **0.026** | **1.3** |

Table E: **Ablation on token's numbers in DisVisioner.** Values marked in orange indicate poor performance, while those in purple represent excessively high scores, which are not desired. Values in green denote the best results, excelling in both editability and ID consistency.

| Method | C-T↑ | C-I↑ | D-I↑ | I-V↓ | mRank↓ |
|---|---|---|---|---|---|
| $n_s = 16, n_i = 16$ | 0.261 | **0.799** | 0.782 | 0.043 | 4.5 |
| $n_s = 4, n_i = 4$ | 0.263 | 0.795 | 0.781 | 0.043 | 4.3 |
| $n_s = 1, n_i = 16$ | 0.312 | 0.353 | 0.341 | 0.021 | 3.3 |
| $n_s = 1, n_i = 4$ | 0.310 | 0.351 | 0.339 | **0.020** | 3.7 |
| $n_s = 16, n_i = 1$ | 0.262 | 0.796 | **0.783** | 0.043 | 4.2 |
| $n_s = 4, n_i = 1$ | 0.268 | 0.794 | 0.780 | 0.041 | 4.0 |
| $n_s = 1, n_i = 1$ **(Ours)** | **0.315** | 0.789 | 0.768 | 0.026 | **3.0** |

semantic and environmental features, it results in slightly higher interval variance (I-V) scores. This indicates that the generated images are influenced by irrelevant factors from the input images.

## F ABLATION ON DISVISIONER AND ENVISIONER

To validate the effectiveness of DisVisioner, we implemented a replacement module, a simple MLP network with fully-connection layers and ReLUs, that eliminates the disentanglement capability of the original DisVisioner, and keeping all other neural network modules unchanged. Specifically, the CLIP *global* image feature, with the shape of `(B, 1, 1024)`, is fed into this MLP, and the output, which has the shape of `(B, 1, 768)`, is passed to EnVisioner. The batch-size `B` is usually set to `1` during inference. As shown in Fig. F, "w/o DisVisioner", whether combined with EnVisioner or not, struggles with editability. This is because irrelevant factors remain entangled with subject-essential factors. When DisVisioner is employed, it effectively disentangles the subject from other factors via feature aggregation in spatial-wise attention layer, resulting in improved editability. As shown in Tab. D, we also computed metrics to quantitatively validate the effectiveness of DisVisioner. Poor editability is reflected in a low C-T score and high C-I, D-I, and I-V values. We also validate the effectiveness of EnVisioner in enhancing details. Both qualitative results in Fig. F and quantitative results in Tab. D demonstrate the advantages and effectiveness of both DisVisioner and EnVisioner.

## G FURTHER ABLATION ON TOKEN NUMBERS OF DISVISIONER

We have quantitatively evaluated the effect of the number of tokens in DisVisioner, as shown in the Tab. E. ① When $n_s \neq n_i$, the image is primarily represented by the larger set without any disentanglement. Specifically, when $n_s > n_i$, the model tends to reconstruct the input image, resulting in lower C-T and higher I-V scores. C-I and D-I scores are also higher due to copying the reference image. When $n_s < n_i$, the subject is not properly represented in the $n_s$ tokens, and the model generates images solely based on the textual description. This leads to higher C-T and lower I-V scores but significantly lower C-I and D-I scores. ② When $n_s = n_i > 1$, the random

initialized subject tokens have more "freedom", capturing most image information and also leading to reconstruction. Under this configuration, the model exhibits lower C-T and higher I-V scores, with increased C-I and D-I scores due to copying the reference. In contrast, ③ when $n_s = n_i = 1$, DisVisioner effectively disentangles the subject from irrelevant factors. This configuration achieves the best C-T, I-V , C-I and D-I scores, where C-I and D-I are not excessively high, as the model avoids simply reconstructing the reference image.

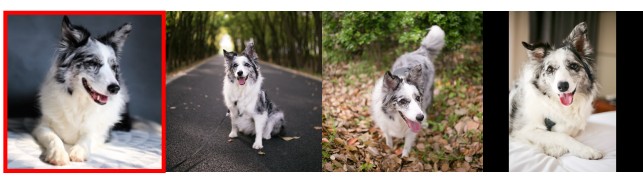
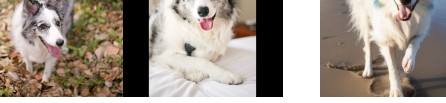
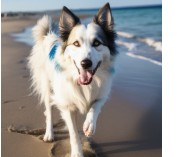
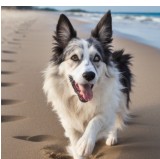

Input images                                          Single Image Inf.    Multiple Images Inf.

Figure H: **Experimental results when using multiple reference images.** These results demonstrate that, when provided with multiple reference images, DisEnvisioner produces slightly higher-quality customizations. All samples are generated using the prompt "a * running on the beach." "Single Image Inf." refers to samples generated under the original setting of DisEnvisioner, while "Multiple Image Inf." denotes the results obtained using the simple multiple-image extension.

## H    EXPERIMENTS WHEN USING MULTIPLE REFERENCE IMAGES

Table F: **Ablation on the effect of multiple reference images**. Best results are **bolded**. For equity, we consider C-I and D-I as two sub-indicators of image-alignment, the rank of each with a weight of **0.5** in the mRank calculation, while the ranks of all other metrics have a weight of **1.0**.

| Method | C-T↑ | C-I↑ | D-I↑ | I-V ↓ | mRank↓ |
|---|---|---|---|---|---|
| DisEnvisioner (Single Image Inf., Ours) | **0.315** | 0.828 | 0.802 | **0.026** | 1.3 |
| DisEnvisioner (Multiply Images Inf.) | **0.315** | **0.831** | **0.803** | **0.026** | **1.0** |

DisEnvisioner can be extended to utilize multiple reference images in a tuning-free manner. This extension is straightforward and does not require re-training the model. Specifically, for a set of images $X = \{x_1, x_2, \ldots, x_n\}$, we first extract their disentangled and enriched subject-essential features $f_i$ using DisVisioner and the projector $P^s$ in EnVisioner, i.e.,

$$f_i = P^s(\text{DisVisioner}(x_i)),$$

where $P^s(\cdot)$ represents the projector in EnVisioner. Next, we compute the average feature $f_{\text{avg}}$ across all subject features as

$$f_{\text{avg}} = \frac{1}{n}\sum_{i=1}^{n} f_i.$$

By injecting this averaged feature into the diffusion model, customized generation based on multiple reference images can be achieved.

As shown in Fig. H and Tab. F, utilizing multiple reference images enhances the quality of subject features, resulting in improved image consistency while maintaining text alignment. This is reflected in the higher C-I and D-I scores compared to using a single image.

## I    QUANTITATIVE COMPARISONS WITH TRANSFORMER-BASED METHODS

As shown in Tab. G, we further compare DisEnvisioner with Transformer-based methods, *i.e.*, SuTI(Chen et al., 2024), Kosmos-G(Pan et al., 2023), and CAFE (Zhou et al., 2024). The results demonstrate that DisEnvisioner still achieves SoTA performance compared to these Transformer-based methods. Notably, the testing datasets for these Transformer-based methods are all based on DreamBooth (Ruiz et al., 2023), and all 30 subjects are used for evaluation, consistent with ours. Therefore, their metrics are directly referenced from their respective papers. We will evaluate their performance of inference time and interval-variance (I-V) for a more comprehensive comparison.

Table G: **Quantitative comparison with existing Transformer-based methods.** The evaluation metrics include text-alignment for assessing editability (C-T), and image-alignment for ID-consistency (C-I, D-I). DisEnvisioner demonstrates better comprehensive performance than other methods. Best results are **bolded** and the second best are underlined. For equity, we consider C-I and D-I as two sub-indicators of image-alignment, the rank of each with a weight of **0.5** in the mRank calculation, while the ranks of all other metrics have a weight of **1.0**.

| Method | C-T↑ | C-I↑ | D-I↑ | mRank↓ |
|---|---|---|---|---|
| Kosmos-G Pan et al. (2023) | 0.287 | **0.847** | 0.694 | 3.3 |
| CAFE (Zhou et al., 2024) | 0.294 | 0.827 | 0.715 | 3.0 |
| SuTI (Chen et al., 2024) | 0.304 | 0.819 | 0.741 | 2.5 |
| **DisEnvisioner (Ours)** | **0.315** | 0.828 | **0.802** | **1.3** |

## J MORE VISUALIZATIONS OF FEATURE DISENTANGLEMENT

As shown in Fig. I, we provide more visualizations of feature disentanglement, across live and non-live subjects. With the $n_s = 1$ and $n_i = 1$, DisVisioner can clearly disentangle subject-essential and -irrelevant features, achieving more accurate image customization quality in diverse scenarios.

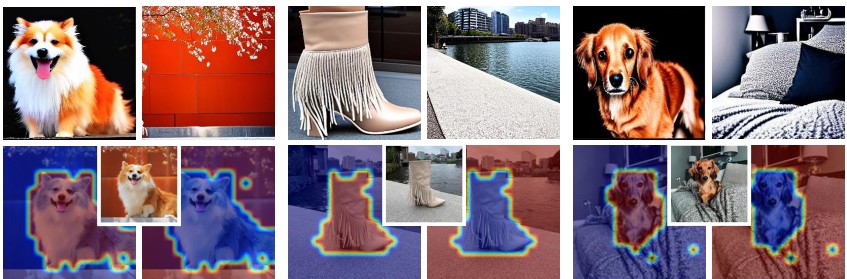

Figure I: **More visualizations of feature disentanglement.** The results showcase DisVisioner's ability to accurately discern subject-essential attributes across diverse scenarios.

## K MORE EXPERIMENTAL DETAILS

### K.1 TRAINING DATA

In the experiments, we utilize the *training set* of OpenImages V6 (Kuznetsova et al., 2020) as the training dataset. Based on this dataset, we construct {prompt, image} pairs for training. As depicted in Fig. J, the training images are derived by cropping and resizing the raw images in accordance with the bounding box annotations. To ensure the quality of the training images, we further filter the cropped images. A cropped image is considered as unsatisfactory and therefore excluded if its area is greater than 80% or less than 2% of the original image's area. As a result, out of 14.61M annotated bounding boxes, we obtain 6.82M {prompt, image} pairs. The text prompts are selected randomly from a CLIP ImageNet template (Radford et al., 2021) and integrated with labelled class names. The complete list of CLIP templates is provided below:

- "a photo of a $S^*$",
- "a rendering of a $S^*$",
- "a cropped photo of the $S^*$",
- "the photo of a $S^*$",
- "a photo of a clean $S^*$",
- "a photo of a dirty $S^*$",
- "a dark photo of the $S^*$",
- "a photo of my $S^*$",

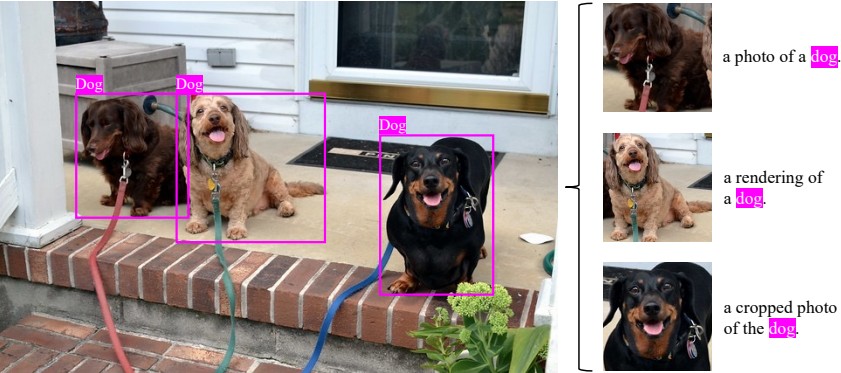

Figure J: **Examples of training data.** The training images are derived by cropping and resizing the raw images in accordance with the bounding box annotations.

- "a photo of the cool $S^*$",
- "a close-up photo of a $S^*$",
- "a bright photo of the $S^*$",
- "a cropped photo of a $S^*$",
- "a photo of the $S^*$",
- "a good photo of the $S^*$",
- "a photo of one $S^*$",
- "a close-up photo of the $S^*$",
- "a rendition of the $S^*$",
- "a photo of the clean $S^*$",
- "a rendition of a $S^*$",
- "a photo of a nice $S^*$",
- "a good photo of a $S^*$",
- "a photo of the nice $S^*$",
- "a photo of the small $S^*$",
- "a photo of the weird $S^*$",
- "a photo of the large $S^*$",
- "a photo of a cool $S^*$",
- "a photo of a small $S^*$"

### K.2 TESTING DATA

For evaluation, we adopt all images and editing prompts from DreamBooth (Ruiz et al., 2023) dataset. It contains a total of 158 images spanning 30 diverse categories, including dog, cat, robot, boot, etc. Fig. K shows a subset of these images. The the complete set of editing prompts for live subjects is detailed below:

- "a $S^*$ in the jungle"
- "a $S^*$ in the snow"
- "a $S^*$ on the beach"
- "a $S^*$ on a cobblestone street"
- "a $S^*$ on top of pink fabric"
- "a $S^*$ on top of a wooden floor"

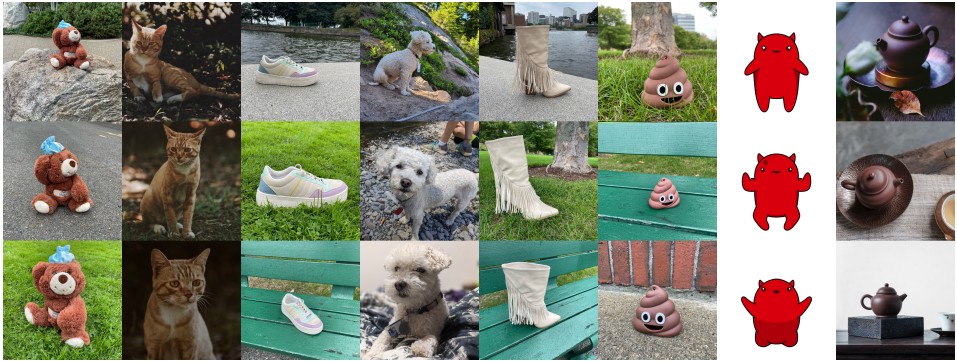

Figure K: **Examples of testing data.** We selected all 30 subjects in the DreamBooth dataset.

- "a $S^*$ with a city in the background"
- "a $S^*$ with a mountain in the background"
- "a $S^*$ with a blue house in the background"
- "a $S^*$ on top of a purple rug in a forest"
- "a $S^*$ with a wheat field in the background"
- "a $S^*$ with a tree and autumn leaves in the background"
- "a $S^*$ with the Eiffel Tower in the background"
- "a $S^*$ floating on top of water"
- "a $S^*$ floating in an ocean of milk"
- "a $S^*$ on top of green grass with sunflowers around it"
- "a $S^*$ on top of a mirror"
- "a $S^*$ on top of the sidewalk in a crowded street"
- "a $S^*$ on top of a dirt road"
- "a $S^*$ on top of a white rug"
- "a red $S^*$"
- "a purple $S^*$"
- "a shiny $S^*$"
- "a wet $S^*$"
- "a cube shaped $S^*$"

And here is the full set of editing prompts for non-live subjects:

- "a $S^*$ in the jungle"
- "a $S^*$ in the snow"
- "a $S^*$ on the beach"
- "a $S^*$ on a cobblestone street"
- "a $S^*$ on top of pink fabric"
- "a $S^*$ on top of a wooden floor"
- "a $S^*$ with a city in the background"
- "a $S^*$ with a mountain in the background"
- "a $S^*$ with a blue house in the background"
- "a $S^*$ on top of a purple rug in a forest"
- "a $S^*$ wearing a red hat"

- "a $S^*$ wearing a Santa hat"
- "a $S^*$ wearing a rainbow scarf"
- "a $S^*$ wearing a black top hat and a monocle"
- "a $S^*$ in a chef outfit"
- "a $S^*$ in a firefighter outfit"
- "a $S^*$ in a police outfit"
- "a $S^*$ wearing pink glasses"
- "a $S^*$ wearing a yellow shirt"
- "a $S^*$ in a purple wizard outfit"
- "a red $S^*$"
- "a purple $S^*$"
- "a shiny $S^*$"
- "a wet $S^*$"
- "a cube shaped $S*$"