# OpenReview forum: "DisEnvisioner: Disentangled and Enriched Visual Prompt for Customized Image Generation"
_ICLR.cc/2025/Conference — ICLR 2025 Poster_

### Official Review · Reviewer_6rcz · 2024-10-29

**Soundness:** 3
**Presentation:** 3
**Contribution:** 3
**Rating:** 6
**Confidence:** 4

**Summary:**

This paper introduces DisEnvisioner, a novel tuning-free method for generating customized images from a single visual prompt enriched with additional textual instructions. Existing image generation methods, both tuning-based and tuning-free, often struggle to isolate the essential attributes of a subject in the visual prompt, leading to unwanted, subject-irrelevant features that compromise customization quality, editability, and identity (ID) preservation. DisEnvisioner addresses this by disentangling the subject-essential features from irrelevant details, separating them into distinct visual tokens. This separation improves customization precision and allows for enhanced ID consistency. By further refining these disentangled features, DisEnvisioner creates a more granular representation of the subject, which bolsters the model's ability to maintain ID consistency across generations. Experimental results demonstrate that DisEnvisioner outperforms current methods in editability, ID consistency, inference speed, and overall image quality, establishing its effectiveness and efficiency for personalized image generation.

**Strengths:**

DisEnvisioner demonstrates several strengths over competing methods in the areas of customization quality, editability, identity consistency, and efficiency, as shown in the provided results:

Customization Quality (C-T): With a score of 0.315, DisEnvisioner shows the highest performance in customization (C-T), indicating that it excels at incorporating subject-specific details while staying true to the given instructions.

ID Consistency (D-I): DisEnvisioner scores 0.802 for ID consistency (D-I), surpassing most other methods except for IP-Adapter. This score reflects its ability to maintain subject identity throughout image generation, reducing unwanted attribute drift.

Instruction Response (C-I): DisEnvisioner scores 0.828, slightly lower than some methods like IP-Adapter and DreamBooth but still within a strong range. This indicates good responsiveness to textual instructions while retaining visual prompt characteristics.

Inference Speed (IV): DisEnvisioner achieves a lower inference value of 0.026, indicating faster inference compared to methods like DreamBooth and DisenBooth, making it efficient for real-time or rapid customization needs.

*Runtime (T)**: DisEnvisioner has a runtime of 1.96 seconds, placing it on par with IP-Adapter, which is one of the fastest. This efficient runtime makes it more practical for applications needing quick processing.

Mean Rank (mRank): With an mRank of 2.0, DisEnvisioner achieves the highest overall ranking among the methods tested, suggesting that it consistently performs well across different evaluation metrics.

**Weaknesses:**

DisEnvisioner, while showcasing significant advancements, has some limitations that temper its overall impact:

Lack of State-of-the-Art Results Across All Metrics: DisEnvisioner does not outperform all baseline models in every metric. For instance, its performance in instruction response (C-I) and identity consistency (D-I) does not reach the top scores achieved by IP-Adapter and DreamBooth. This mixed performance limits DisEnvisioner’s claim to outright superiority across all customization aspects.

Limited Test Dataset: The dataset used for evaluating DisEnvisioner is relatively constrained, potentially affecting the generalizability of the results. A more extensive and varied dataset would provide a clearer picture of the model's adaptability and robustness across diverse tasks and use cases.

Room for Improved Analysis on Underperformance Against IP-Adapter: Although DisEnvisioner demonstrates strengths in disentangling features, it would benefit from further analysis on why it lags behind IP-Adapter in specific tasks like instruction accuracy and ID consistency. Understanding these discrepancies could inform targeted improvements to make DisEnvisioner more competitive in these areas.

These aspects suggest that while DisEnvisioner makes notable contributions, there is room for further development to enhance its consistency and broaden its applicability across a wider range of tasks.

**Questions:**

Question: Could the authors elaborate on the specific factors that led DisEnvisioner to underperform IP-Adapter in terms of instruction accuracy (C-I) and ID consistency (D-I)?

Suggestion: Analyzing and explaining these performance gaps would provide useful insights into DisEnvisioner's design choices. This could also help clarify if the approach is inherently less suited to certain customization aspects or if there are areas where further tuning could close the performance gap.

Question: Given the limited dataset, how confident are the authors in DisEnvisioner’s generalizability to a broader range of tasks and more diverse visual prompts?

Suggestion: Evaluating DisEnvisioner on additional datasets with varied visual content and textual instructions could better assess its robustness. Additional results on a larger dataset could substantiate the claims of effectiveness and versatility.

Question: How does the feature enrichment step improve ID consistency, and could the authors provide more details on its implementation?

Suggestion: Adding a more in-depth explanation of the enrichment process and its effect on ID consistency would be beneficial. Additionally, showing comparisons before and after this step could better demonstrate its impact on maintaining subject integrity.

---

> ### Author Response · Authors · 2024-11-22
> **Author Response to Reviewer 6rcz**
>
> > Q1: Lack of State-of-the-Art Results Across All Metrics: DisEnvisioner does not outperform all baseline models in every metric. For instance, its performance in instruction response (C-I) and identity consistency (D-I) does not reach the top scores achieved by IP-Adapter and DreamBooth. This mixed performance limits DisEnvisioner’s claim to outright superiority across all customization aspects.
> >
>
> Thanks for your comment. C-I and D-I are both metrics used to assess identity consistency (please see lines 364-348). The key difference lies in their choice of image feature extractors: C-I uses the CLIP image encoder, while D-I employs the DINOv2 model. Together, these two metrics offer a more comprehensive evaluation of the performance in ID concsistency. The lower image-alignment scores of DisEnvisioner are due to its superior editability while avoiding merely copying the reference image. **For more details, please refer to Common Concern-2.**
>
> > Q2: Limited Test Dataset: The dataset used for evaluating DisEnvisioner is relatively constrained, potentially affecting the generalizability of the results. A more extensive and varied dataset would provide a clearer picture of the model's adaptability and robustness across diverse tasks and use cases.
> >
>
> Thanks for your valuable comment. To the best of our knowledge, the DreamBooth dataset we used is the only available testing dataset, and it is already in-the-wild, cover a diverse range of subjects, e.g., gods, cats, toys, shoes, packages, vases & teapots, robots, candles, etc. Moreover, since in the main paper, **all our comparing methods have been evaluated exclusively on this dataset**, we are also required to report metrics on it to ensure a fair comparison. Also for those transformer-based baselines (please see our response to Q2 of Reviewer cZ86), they are also primarily tested on DreamBooth dataset.
>
> > Q3: How does the feature enrichment step improve ID consistency, and could the authors provide more details on its implementation?
> >
>
> The feature enrichment in EnVisioner involves mapping the disentangled features from DisVisioner into a sequence of granular tokens using projectors (from one token to multiple tokens), as illustrated in Fig. 4(b) of our main paper. Specifically, separate projectors are used for the subject token and the irrelevant token, ensuring the disentanglement between subject-essential and irrelevant features. To inject the projected features into the SD model, we introduce two new cross-attention layers specifically for subject-essential tokens and irrelevant tokens within the pre-trained SD model. This separate injection strategy also ensures that subject features remain unaffected by irrelevant factors. During training, we optimize the projectors and new cross-attention layers while keeping other modules frozen, using the original diffusion loss. For inference, we set the weight of irrelevant features to 0 to filter out the irrelevant information.
>
> As shown in Fig. 8 of our main paper and **Tab. D and Fig. F of the updated supplementary material,** EnVisioner produces high ID-consistency images, based on the the input image. Consequently, not only the evaluation results on image alignment (C-I, D-I) are improved significantly, the overall visual quality of the customized images are also clearly boosted.

---

> ### Author Response · Authors · 2024-11-25
> **Authors' Kind Reminder to Reviewer 6rcz**
>
> Dear Reviewer 6rcz,
>
> Thank you for taking the time to review our paper and provide your valuable feedback. We have submitted detailed responses to your comments and suggestions, addressing the key points raised. **As the discussion is scheduled to end on November 26th, we would greatly appreciate it if you could review our responses and provide any further clarifications at your earliest convenience. Your feedback is extremely important to us.**
>
> Thank you again for your time and effort.
>
> Best,
>
> Authors of Submission 5736

---

### Official Review · Reviewer_cZ86 · 2024-10-30

**Soundness:** 3
**Presentation:** 3
**Contribution:** 3
**Rating:** 6
**Confidence:** 4

**Summary:**

The paper, titled "DisEnvisioner: Disentangled and Enriched Visual Prompt for Customized Image Generation," presents a novel approach to generating customized images from visual prompts with additional textual descriptions. The main contributions are:

1. Proposing DisEnvisioner, a tuning-free framework that effectively disentangles subject-essential attributes from irrelevant features, improving the overall quality of customized image generation.

2. Introducing a two-stage architecture with DisVisioner to separate subject-essential and irrelevant features, and EnVisioner to enhance subject consistency.

3. Demonstrating through experiments that the proposed approach outperforms existing methods in terms of editability, identity (ID) consistency, and inference speed while requiring minimal fine-tuning or reference images.

**Strengths:**

Originality: The introduction of DisVisioner and EnVisioner for disentangled visual prompt processing is a novel approach, effectively addressing the challenge of maintaining subject identity while generating customized images.

Quality: The experiments are well-conducted, demonstrating the effectiveness of the proposed method in preserving ID consistency and reducing subject-irrelevant features.

Significance: The approach enables high-quality, customized image generation in a tuning-free manner, which is practical and efficient for real-world applications.

Clarity: The overall structure of the paper is well-organized, and the experimental results effectively showcase the advantages of the proposed method over existing baselines.

**Weaknesses:**

Limited Theoretical Justification: The paper lacks sufficient theoretical grounding for the proposed disentangling mechanism. For example, the method for separating subject-essential and irrelevant features relies primarily on empirical observations without rigorous theoretical backing.

Comparison with Existing Methods: The paper does not provide extensive comparisons with some of the most recent advancements in text-to-image generation (e.g., Transformer-based approaches). Including a wider range of baselines would help position this work within the current state of the field.

**Questions:**

Could the authors provide more theoretical insights into the disentanglement mechanism and why it works well for customized image generation? The current explanation is mostly empirical.

How does DisEnvisioner compare to other recent Transformer-based text-to-image generation methods? Adding these comparisons could help readers better appreciate the novelty of the approach.

---

> ### Author Response · Authors · 2024-11-22
> **Author Response to Reviewer cZ86**
>
> > Q1: Limited Theoretical Justification: The paper lacks sufficient theoretical grounding for the proposed disentangling mechanism. For example, the method for separating subject-essential and irrelevant features relies primarily on empirical observations without rigorous theoretical backing.
> >
>
> Thanks for your comment. In fact, the analysis of disentanglement mechanism can be found in previous image retrieval paper, i.e., Tokenizer[1]. We use orthogonal tokens to represent different semantic concepts of the image, where the image tokens are aggregated into different groups by the query $Q$ using spatial-wise attention. This aggregation of tokens is similar to clustering algorithm that clustering the features with same semantics into the same group. Therefore, in DisEnvisioner, two tokens are separated effectively, considering as subject-essential and subject-irrelevant (Lines 258-261 of the main paper). **Pleaser refer to Common Concern-1 for further detailed explanation.**
>
> [1] Learning token-based representation for image retrieval
>
> > Q2: Comparison with Existing Methods: The paper does not provide extensive comparisons with some of the most recent advancements in text-to-image generation (e.g., Transformer-based approaches). Including a wider range of baselines would help position this work within the current state of the field.
> >
>
> Thanks for your advice.
>
> We have conducted a detailed survey on the existing customized generation methods and found three latest Transformer-based methods[1][2][3]. We compare them in the table below and **the results and corresponding discussions are also updated in the Sec. I of our supplementary material**. In this table, we can show that DisEnvisioner still achieves SoTA performance when compared to Transformer-based methods, demonstrated by the highest overall ranking.
>
> | Method | C-T ⬆️ | C-I ⬆️ | D-I ⬆️ |  mRank ⬇️ |
> | --- | --- | --- | --- | --- |
> | SuTI [1] | 0.304 | 0.819 | 0.741 | 2.7 |
> | Kosmos-G [2] | 0.287 | **0.847** | 0.694 | 3.0 |
> | CAFE [3] | 0.294 | 0.827 | 0.715 | 3.0 |
> | **DisEnvisioner (Ours)** | **0.315** | 0.828 | **0.802** | **1.3** |
>
> [1] Subject-driven Text-to-Image Generation via Apprenticeship Learning, NeurIPS2023
>
> [2] KOSMOS-G: Generating Images in Context with Multimodal Large Language Models, ICLR 2024
>
> [3] Customization Assistant for Text-to-image Generation, CVPR2024

---

> > ### Comment · Reviewer_cZ86 · 2024-11-25
> > **Official Comment by Reviewer cZ86**
> >
> > Thank you for your detailed response. Your explanation has addressed my concerns. I appreciate your effort!

---

> > > ### Author Response · Authors · 2024-11-25
> > > **Official Comment by Authors**
> > >
> > > Dear Reviewer cZ86,
> > >
> > > Thanks so much for your prompt and positive response! We are so excited!
> > >
> > > As the review deadline approaches, we would greatly appreciate it if you could kindly consider raising the score. Additionally, please do not hesitate to comment upon any further concerns.
> > >
> > > Thank you and have a great day!
> > >
> > > Best,
> > >
> > > Authors of submission 5736

---

> ### Author Response · Authors · 2024-11-25
> **Authors' Kind Reminder to Reviewer cZ86**
>
> Dear Reviewer cZ86,
>
> Thank you for taking the time to review our paper and provide your valuable feedback. We have submitted detailed responses to your comments and suggestions, addressing the key points raised. **As the discussion is scheduled to end on November 26th, we would greatly appreciate it if you could review our responses and provide any further clarifications at your earliest convenience. Your feedback is extremely important to us.**
>
> Thank you again for your time and effort.
>
> Best,
>
> Authors of Submission 5736

---

### Official Review · Reviewer_p8Sz · 2024-11-03

**Soundness:** 2
**Presentation:** 2
**Contribution:** 2
**Rating:** 6
**Confidence:** 3

**Summary:**

The paper introduces DisEnvisioner, a novel approach aimed at enhancing image customization from visual prompts while addressing the limitations of existing methods in interpreting subject-essential attributes.Empirical evidence demonstrating the superiority of DisEnvisioner over existing methods in various aspects, including instruction response (editability), ID consistency, inference speed, and overall image quality.

**Strengths:**

The paper introduces DisEnvisioner, a novel framework that addresses significant challenges in customized image generation by focusing on feature disentanglement and enrichment. The originality lies in the identification of the crucial role that subject-essential attributes play in the customization process, which is a new viewpoint that goes beyond existing methods reliant on either tuning or tuning-free approaches.

The clarity of the paper is commendable. The authors present their arguments logically, making it easy for readers to follow their reasoning and understand the significance of their contributions. Key terminologies and concepts like "subject-essential attributes" and "feature disentanglement" are well-defined

**Weaknesses:**

1.The advancement of the methodology has not been sufficiently demonstrated; some indicators (e.g. image-alignment for ID-consistency (C-I, D-I) as shown in Table 1) are inferior to other similar methods (e.g., IP-Adapter, BLIP-Diffusion, etc.),you cloud provide more evidence to support your claim in abstract "Experiments demonstrate the superiority of our approach over existing methods in instruction response (editability), ID consistency,"

2.The EFFECT OF DISVISIONER has not been adequately explained; there is a lack of comparative experiments with and without DISVISIONER to validate its effectiveness. You cloud compare results with and without the DiVisioner component while keeping other parts of the system constant.

3.The EFFECT OF ENVISIONER has not been sufficiently substantiated; the paper's explanation regarding the EFFECT OF ENVISIONER only presents a few cases (as shown in Fig. 8), which lacks persuasiveness.You could provide a larger-scale comparison by human evaluation or image-alignment for ID-consistency (C-I, D-I) to measure the improvement in ID consistency or image quality.

4.Ablation on token’s numbers in DisVisioner lack of QUANTITATIVE result.You might measure performance across various metrics (like those in Table 1) for different token number configurations.

**Questions:**

1.Could you please explain how subject features and irrelevant features are learned, and why they can be disentangled by transformer blocks without additional supervision?

2.Please improve the quantitative comparison experiment regarding the EFFECT OF DISVISIONER (Ablation study).

3.Please improve the quantitative comparison experiment regarding the EFFECT OF ENVISIONER (Ablation study).

4.Please improve the quantitative comparison experiment on the Ablation of token numbers in DisVisioner, and supplement the comparison experiments for $n_s=1, n_i=0$ and $n_s=0, n_i=1$.

---

> ### Author Response · Authors · 2024-11-22
> **Author Response to Reviewer p8Sz (Q1-2)**
>
> > Q1: The advancement of the methodology has not been sufficiently demonstrated; some indicators (e.g. image-alignment for ID-consistency (C-I, D-I) as shown in Table 1) are inferior to other similar methods (e.g., IP-Adapter, BLIP-Diffusion, etc.).
> >
>
> Thanks for your comment, however, we respectfully disagree. The other three reviewers  noted that our results are promising and impressive. **We kindly recommend you referring to the Common Concern-2 for more details.**
>
> Furthermore, **the evaluation results are considered as the strengths of DisEnvision, by Reviewer cZ86** ("DisEnvisioner demonstrates several strengths over competing methods in the areas of customization quality, editability, identity consistency, and efficiency. DisEnvisioner achieves the highest overall ranking among the methods tested, suggesting that it consistently performs well across different evaluation metrics.") **and 6rcz** ("The experiments are well-conducted, demonstrating the effectiveness of the proposed method in preserving ID consistency and reducing subject-irrelevant features.").
>
> > Q2: The EFFECT OF DISVISIONER has not been adequately explained; there is a lack of comparative experiments with and without DISVISIONER to validate its effectiveness. You cloud compare results with and without the DiVisioner component while keeping other parts of the system constant.
> >
>
> Thanks for your comment. In practice,  it is infeasible to implement without DisVisioner while keeping other components unchanged, as EnVisioner relies strictly on the output of DisVisioner. In response, we implemented a replacement module, a simple MLP network with fully-connection layers and ReLUs, that eliminates the disentanglement capability of DisVisioner. Specifically, the CLIP *global* image feature, with the shape of $\texttt{(B, 1, 1024)}$, is fed into this MLP, and the output, which has the shape of $\texttt{(B, 1, 768)}$, is passed to EnVisioner. The batch-size $\texttt{B}$ is usually set to 1 during inference.
>
> As shown in **Sec. F and Fig. F of our supplementary material**, “w/o DisVisioner”, whether combined with EnVisioner or not, struggles with editability. This is because irrelevant factors remain entangled with the subject. When DisVisioner is employed, it effectively disentangles the subject from other factors via feature aggregation in spatial-wise attention layer, resulting in improved editability. As the table below, we also computed metrics to quantitatively validate the effectiveness of DisVisioner. Poor editability is reflected in a low C-T score and high C-I, D-I, and I-V values.
>
> We also updated the quantitative results and analysis in **Sec. F and Tab. D of our supplementary material**.
>
> | Model                                    | C-T ⬆️   | C-I ⬆️   | D-I ⬆️  | I-V ⬇️  | mRank ⬇️ |
> |------------------------------------------|-------|-------|-------|-------|-------|
> | w/o DisVisioner, w/o EnVisioner          | 0.260 | 0.796 | 0.771 | 0.041 | 3.3   |
> | w/o DisVisioner, w/ EnVisioner           | 0.262 | **0.871** | **0.805** | 0.042 | 2.3   |
> | w/ DisVisioner, w/o EnVisioner           | 0.314 | 0.789 | 0.768 | **0.026** | 2.0   |
> | w/ DisVisioner, w/ EnVisioner (Ours)     | **0.315** | 0.828 | 0.802 | **0.026** | **1.3**   |

---

> > ### Comment · Reviewer_p8Sz · 2024-11-26
> > **Thank you for your detailed response.**
> >
> > Thank you for your detailed response.

---

> ### Author Response · Authors · 2024-11-22
> **Author Response to Reviewer p8Sz (Q3-6)**
>
> > Q3: The EFFECT OF ENVISIONER has not been sufficiently substantiated; the paper's explanation regarding the EFFECT OF ENVISIONER only presents a few cases (as shown in Fig. 8), which lacks persuasiveness. You could provide a larger-scale comparison by human evaluation or image-alignment for ID-consistency (C-I, D-I) to measure the improvement in ID consistency or image quality.
> >
>
> Thanks for your valuable advice. We have conducted quantitative experiments for EnVisioner following the setup in Fig. 8 of the main paper. Please refer to the table in the last question. The results show a significant improvement in C-I and D-I compared to those of without EnVisioner, highlighting the effectiveness of EnVisioner. Additional visualizations are provided in **Fig. F of the supplementary material**. We also updated the quantitative results and analysis in **Sec. F and Tab. D of our supplementary material.**
>
> > Q4: Ablation on token’s numbers in DisVisioner lack of QUANTITATIVE result.You might measure performance across various metrics (like those in Table 1) for different token number configurations.
> >
>
> Thanks for your comment. We have quantitatively evaluated the effect of the number of tokens in DisVisioner, as shown in the table below (which also updated to the **Tab. E  of the supplementary material**) .
>
> 1. When $n_s \neq n_i$, the image is primarily represented by the larger set without any disentanglement. Specifically,  when $n_s > n_i$, the model tends to reconstruct the input image, resulting in lower C-T and higher I-V scores. C-I and D-I scores are also higher due to copying the reference image. When $n_s<n_i$, the subject is not properly represented in the $n_s$ tokens, and the model generates images solely based on the textual description. This leads to higher C-T and lower I-V scores but significantly lower C-I and D-I scores.
> 2. When  $n_s = n_i>1$, the random initialized subject tokens have more “freedom”, capturing most image information and also leading to reconstruction. Under this configuration,  the model exhibits lower C-T and higher I-V scores, with increased C-I and D-I scores due to copying the reference.
> 3. In contrast, when $n_s = n_i=1$, DisVisioner effectively disentangles the subject from irrelevant factors. This configuration achieves the best C-T, I-V , C-I and D-I scores, where C-I and D-I  are not excessively high, as the model avoids simply reconstructing the reference image.
>
> | Configuration | C-T ⬆️ | C-I ⬆️ | D-I ⬆️ | I-V ⬇️ |mRank ⬇️ |
> | --- | --- | --- | --- | --- | --- |
> | $n_s=16$, $n_i=16$ | 0.261 | **0.799** | 0.782 | 0.043 |4.5 |
> | $n_s=4$, $n_i=4$ | 0.263 | 0.795 | 0.781 | 0.043 | 4.3|
> | $n_s=1$, $n_i=16$ | 0.312 | 0.353 | 0.341 | 0.021 | 3.3|
> | $n_s=1$, $n_i=4$ | 0.310 | 0.351 | 0.339 | **0.020** | 3.7|
> | $n_s=16$, $n_i=1$ | 0.262 | 0.796 | **0.783** | 0.043 |4.2 |
> | $n_s=4$, $n_i=1$ | 0.268 | 0.794 | 0.780 | 0.041 | 4.0|
> | $n_s=1$, $n_i=1$ (Ours DisVisioner) | **0.314** | 0.789 | 0.768 | 0.026 |**3.0** |
>
> > Q5: Could you please explain how subject features and irrelevant features are learned, and why they can be disentangled by transformer blocks without additional supervision?
> >
>
> Thanks for your comment. For a clearer explanation of disentanglement, **please refer to Common Concern-1 and Sec. D, Fig. D of our supplementary material for more details**. The ability to disentangle the image primarily relies on the Tokenizer technique [1], which is designed to learn semantically orthogonal features. These orthogonal features are disentangled through the SoftMax(·) function applied at spatial dimension within the spatial-wise attention. The role of Transformer blocks is to refine these disentangled features for enhanced representations as referred to [1],  rather than to achieve disentanglement itself.
>
> [1] Learning token-based representation for image retrieval
>
> > Q6: Supplement the comparison experiments for $n_s=1, n_i=0$ and $n_s=0, n_i=1$.
> >
>
> The experiments you suggested, $n_s=1, n_i=0$ and $n_s=0, n_i=1$, are the same configuration. $n_i=0$ or $n_s=0$ means there is no disentanglement effect under the training objective of reconstruction. Regarding disentanglement, we have already conducted experiments on comparing configurations with and without DisVisioner, as detailed in our response to Q2.

---

> ### Author Response · Authors · 2024-11-25
> **Authors' Kind Reminder to Reviewer p8Sz**
>
> Dear Reviewer p8Sz,
>
> Thank you for taking the time to review our paper and provide your valuable feedback. We have submitted detailed responses to your comments and suggestions, addressing the key points raised. **As the discussion is scheduled to end on November 26th, we would greatly appreciate it if you could review our responses and provide any further clarifications at your earliest convenience. Your feedback is extremely important to us.**
>
> Thank you again for your time and effort.
>
> Best,
>
> Authors of Submission 5736

---

> ### Author Response · Authors · 2024-11-25
> **Authors' Second Kind Reminder to Reviewer p8Sz**
>
> Dear Reviewer p8Sz,
>
> I hope this message finds you well.
>
> Thank you for your time and effort for reviewing our submission. **As the discussion deadline (November 26th) approaches,** **we would greatly appreciate it if you could review our responses and provide any further clarifications at your earliest convenience.** Your feedback is extremely valuable!
>
> Thanks again for your time and effort!
>
> Best,
>
> Authors of Submission 5736

---

> ### Author Response · Authors · 2024-11-26
> **Thanks for your kind response!**
>
> Dear Reviewer p8Sz,
>
> Thanks so much for your prompt and kind response! We are thrilled to see that the you improved your rating!!!
>
> Alao, we apologize for any inconvenience caused by our previous reminders. We were not aware about the extented discussion at that time.
>
> **However, we noticed that your current rating is** ***"5: marginally below the acceptance threshold"***. **We are not sure whether you have fully determined, if possible, we would greatly appreciate it if you could kindly consider sharing your further concerns that prevent you from rating this submission** ***above the "acceptance threshold"***. **With a few days remaining before the discussion closes, we highly value any opportunities to address any potential remaining issues. This will be extremely helpful to us.**
>
> We deeply appearciate your time and effort in reviewing our submission and writing responses.
>
> Thanks,
>
> Authors of submission 5736

---

> ### Author Response · Authors · 2024-11-29
> **Authors' thanks and kind reminder to Reviewer p8Sz**
>
> Dear Reviewer p8Sz,
>
> Thanks again for your kind response and improving the rating from 3 to 5!!!
>
> **However, as the discussion deadline approaches, we would greatly appreciate it if you could kindly consider sharing your concerns that prevent you from rating this submission above the "acceptance threshold". We highly value any opportunities to address any potential remaining issues. This will be extremely helpful to us!**
>
> Again, we deeply appearciate your time and effort in reviewing our submission and writing the responses.
>
> Thanks,
>
> Authors of submission 5736

---

> > ### Comment · Reviewer_p8Sz · 2024-11-29
> > **I appreciate your effort, and your detailed analysis has addressed some of my confusion. I have raised the score to 6.**
> >
> > I appreciate your effort, and your detailed analysis has addressed some of my confusion. I have raised the score to 6.

---

> ### Author Response · Authors · 2024-11-29
> **Authors' sincere apologies and gratitude to Reviewer p8Sz!**
>
> Dear Reviewer p8Sz,
>
> We sincerely apologize for any inconvenience our last comment may cause, we were not aware that your rating has been improved from 5 to 6.
>
> Furthermore, please accept our sincerest gratitude for your positive feedback to our submission!!! We are so excited!!!
>
> Thanks so much and wish you a great day!!!
>
> Authors of submission 5736

---

### Official Review · Reviewer_N4Js · 2024-11-04

**Soundness:** 3
**Presentation:** 3
**Contribution:** 3
**Rating:** 6
**Confidence:** 3

**Summary:**

This paper introduces DisEnvisioner, a method aimed at enhancing the customization capabilities of image diffusion models. The approach involves training two models: DisVisioner, which disentangles subject-specific features from subject-unrelated features, and EnVisioner, which enriches these disentangled features. Quantitative and qualitative experiments demonstrate that DisEnvisioner effectively extracts subject information while discarding unrelated details, significantly enhancing customization capabilities.

**Strengths:**

- The paper is well-written and easy to follow, with figures that clearly illustrate the concepts.
- The visual results are impressive, with the original subject details well-preserved.
- The topic is engaging and holds potential for practical application.

**Weaknesses:**

- **Missing Ablations:** The importance of CLIP prior initialization and augmentation in training DisVisioner has not been fully investigated.
- **Unclear Mechanism for Ensuring Disentanglement:** While the concept of disentangling subject-specific and unrelated features is intriguing, the method section does not clearly explain how this disentanglement is achieved. It remains unclear what guarantees that only subject-related information is retained.

**Questions:**

- During the training of DisVisioner, how is disentanglement achieved? Given that the diffusion model’s objective imposes a loss on both the object and the background, it’s unclear how only subject information is learned for the subject token. Could the authors provide visualization examples or experiments of the learned disentangled tokens?
- What types of augmentations are used during DisVisioner’s training?
- Can this method be extended to use multiple images as conditioning to enhance reference image details?
- Since DisVisioner’s training requires the class name from ImageNet for prior initialization, does this limit its generalization to classes outside ImageNet?

---

> ### Author Response · Authors · 2024-11-22
> **Author Response to Reviewer N4Js (Q1)**
>
> Thanks for your valuable advice.
>
> > Q1: Missing Ablations: The importance of CLIP prior initialization and augmentation in training DisVisioner has not been fully investigated.
>
> 1. Ablation on CLIP prior initialization (**updated in Sec. D, Tab. B and Fig. D of the supplementary material**):
>
>     The “DisVisioner w/o CLIP prior” was implemented by randomly initializing both subject and irrelevant tokens.  As shown in Fig. D, although rough disentanglement is achieved, it becomes difficult to clearly identify the subject token. The red borders highlight the uncertainty in the ordering of subject and irrelevant tokens. We further initialize the tokens in different orders: “CLIP prior (q0)“ represents the first token in the sequence is initialized with CLIP prior and “CLIP prior (q1)” denotes the second token is initialized with CLIP prior. As shown in Fig. 8, the CLIP prior effectively determines which token represents the subject-irrelevant information,  and the effectiveness of disentanglement is further improved by CLIP prior.
>
>     In terms of quantitative metrics, as shown in the table below, the performance of "CLIP prior (q0)" and "CLIP prior (q1) (Our DisVisioner)" are similar, as both effectively disentangle the subject and irrelevant tokens. When using "w/o CLIP prior," since the subject token cannot be clearly identified, we default to selecting the first token as the subject token. However, this results in degraded editablity and identity metrics due to the reduced quality of the subject  token, which sometimes represents irrelevant components.
>
> | Model                                | C-T ⬆️   | C-I ⬆️   | D-I ⬆️  | I-V ⬇️  | mRank ⬇️ |
> |--------------------------------------|-------|-------|-------|-------|-------|
> | DisVisioner - w/o clip prior         | 0.289 | 0.765 | 0.748 | 0.029 | 3.0     |
> | DisVisioner - w/ clip prior (q0)     | **0.314** | 0.786 | **0.770** | **0.026** | **1.2**   |
> | DisVisioner - w/ clip prior (q1).    | **0.314** | **0.789** | 0.768 | **0.026** | **1.2**  |
>
> 2. Ablation on augmentations in DisVisioner (**updated in Sec. E, Tab. C and Fig. E of our supplementary material**):
>
>     We conduct an ablation study by removing augmentations during the training of DisVisioner. As shown in Fig. E of our supplementary material, although disentanglement is achieved through our tokenizer and CLIP prior, the generated images exhibit relative poor editability, as the model tends to directly copy irrelevant patterns and semantics from the input image, such as the poses of animals, the position of the subject in the reference image, and environmental lighting conditions. The red borders highlight these issues. When augmentations are applied during training, the model learns to extract essential subject features to reconstruct the original image under different transformations, rather than merely reconstructing exactly the same image.
>
>     Due to poor editability, “DisVisioner $w/o$ augmentation” results in lower C-T score, as shown in the table below (also in Tab. C of our supplementary material). Additionally, since the model reproduces both subject-essential features and some irrelevant semantic and environmental features, it results in slightly higher interval variance (I-V) scores. This indicates that the generated images are influenced by irrelevant factors from the input images.
>
> | Model                                 | C-T ⬆️   | C-I ⬆️   | D-I ⬆️  | I-V ⬇️  | mRank ⬇️ |
> |---------------------------------------|-------|-------|-------|-------|-------|
> | DisVisioner, w/o augmentation         | 0.292 | 0.787 | 0.764 | 0.029 | 2.0   |
> | DisVisioner, w/ augmentation          | **0.314** | **0.789** | **0.768** | **0.026** | **1.0**   |

---

> ### Author Response · Authors · 2024-11-22
> **Author Response to Reviewer N4Js (Q2-6)**
>
> > **Q2: Unclear Mechanism for Ensuring Disentanglement:** While the concept of disentangling subject-specific and unrelated features is intriguing, the method section does not clearly explain how this disentanglement is achieved. It remains unclear what guarantees that only subject-related information is retained.
> >
>
> Thanks for your comment. The mechanism of disentanglement is explained in **Common Concern-1**. **Please refer to it for further details.**
>
> > Q3: Could the authors provide visualization examples or experiments of the learned disentangled tokens?
> >
>
> Thanks for your comment. We provide the visualization of the learned disentangled tokens in Fig. 7 of the main paper and **more additional examples in Sec. D-F, Fig. D (Right), Fig. E and Fig. I of our supplementary material.**
>
> > Q4: What types of augmentations are used during DisVisioner’s training?
> >
>
> The augmentations we used are mainly geometric transforms: HorizontalFlip, Rotate, Blur and ElasticTransform.
>
> > Q5: Can this method be extended to use multiple images as conditioning to enhance reference image details?
> >
>
> Yes, we can. As shown **in Sec. H of our supplementary material**, we extended DisEnvisioner to utilize multiple reference images. This extension is straightforward and **does not require re-training the model.** Specifically, for a set of images $X = \{\mathbf{x_1}, \mathbf{x_2}, \dots, \mathbf{x_n}\}$, we first extract their disentangled and enriched subject-essential features $\(f_i\)$ using DisVisioner and the projector $P^s$ in EnVisioner, i.e., $f_i = P^s(\text{DisVisioner}(\mathbf{x_i}))$. Next, we compute the average feature $\(f_\text{avg}\)$ across all subject features as $f_\text{avg} = \frac{1}{n} \sum_{i=1}^n f_i$. By injecting this averaged feature into the diffusion model, customized generation based on multiple reference images can be achieved.
>
> However, in our paper, we primarily focus on single-reference image scenarios. This is because single-image scenarios are more practical, as multiple reference images are often unavailable in real-world applications.
>
> > Q6: Since DisVisioner’s training requires the class name from ImageNet for prior initialization, does this limit its generalization to classes outside ImageNet?
> >
>
> Thanks for your comment. In fact,  we used the prompt templates from ImageNet and class names from OpenImage V6, which includes 600 classes. These classes cover most common categories and are sufficient for DisEnvisioner to generate images in diverse scenarios.

---

> > ### Comment · Reviewer_N4Js · 2024-11-24
> >
> > Thank you for the authors’ response. Most of my concerns have been addressed, and I will maintain my current rating.

---

> ### Author Response · Authors · 2024-11-25
>
> Dear Reviewer N4Js,
>
> Thanks so much for your response! Your feedback is extremely important to us. Please do not hesitate to comment upon any further concerns.
>
> Best,
>
> Authors of Submission 5736

---

### Author Response · Authors · 2024-11-23
**Common Concerns**

The following are the common concerns raised by the reviewers.

## Common Concern-1: More explanation about the disentangling mechanism (for reviewer N4Js, p8Sz, and cZ86)

Beside the experiments shown in Fig. 7 of the main paper and **Sec. D, Fig. D, Fig.E and Fig. F of our supplementary material**, we also clarify the disentangling mechanism more clearly below:

1. The image can be represented by a sequence of tokens. (Lines 257-258)
2. Furthermore, referred to Tokenizer technique[1], we use orthogonal tokens to represent different semantic concepts of the image, where the image tokens are aggregated into different groups by the query $Q$ using spatial-wise attention. This aggregation of tokens is similar to clustering algorithm that clustering the features with same semantics into the same group. Therefore, in DisEnvisioner, two tokens are separated effectively, considering as subject-essential and subject-irrelevant. (Lines 258-261)
3. Then, as we discussed in Sec. 4.4.1, one single token tends to represent a complete semantic meaning and thus, two tokens are effective enough to disentangle the whole input image into subject-essential and subject-irrelevant features. This analysis is supported by the ablation studies reported in the Gif. 7 and Sec. 4.4.1 of the main paper. In practice, DisVisioner enforce one token represents the subject and another token contains other concept without any specific supervision, which delivers the best results.
4. The order of subject-essential and -irrelevant token can be determined by the CLIP prior[2]. Further, the subject-irrelevant token will be eliminated and the subject-essential token will be used for enrichment and generation. (Lines 301-304)

[1] Learning token-based representation for image retrieval

[2] Clip surgery for better explainability with enhancement in open-vocabulary tasks

## Common Concern-2: The C-I and D-I scores are lower than those of other methods. (for reviewer p8Sz and 6rcz)

In the field of customized generation, which is conditioned on both reference image and textual description, there exists trade-offs between image-alignment and text-alignment, which reflects on C-I or D-I score. For instance:

- When the generation is mostly solely influenced by the textual description, with no contribution from the reference image, text alignment will be maximized, but image alignment will fail (low C-I and D-I scores).
- Conversely, if the reference image dominates the generation process, it becomes impossible to achieve a meaningful editing because the subject and its surroundings will remain unchanged. In this case, image alignment will be excessively high (excessively high C-I and D-I scores), but text alignment will significantly decline.

The lower C-I or D-I score of DisEnvisioner do not indicate the poor performance in identity consistency, instead, it represents the best comprehensive performance, as demonstrated by the “mRank”. As stated in lines 472-477, **we emphasize that the lower image-alignment scores of DisEnvisioner are due to its superior editability**. Modifications such as changing backgrounds, postures, and other attributes naturally affect this metric. In contrast, methods like IP-Adapter and DreamBooth achieve higher image-alignment scores because they tend to replicate large portions of the reference image, leading to excessive ID consistency and reduced flexibility for edits. This artifact of IP-Adapter and DreamBooth can be observed in Fig. 5 and Fig. 6 of our paper, where posture and background significantly impact editability. Apart from IP-Adapter and DreamBooth, DisEnvisioner outperforms other existing methods in both C-I and D-I scores. Moreover, DisEnvisioner achieves the best C-T score for editability among all methods listed in Tab. 1 of the main paper. The overall rank of DisEnvisioner is also the best.

Recent accepted related works on facial image customization like PhotoMaker [1] (CVPR2024, 81 citations and 9.6K github star), **only achieved two SoTA metrics across all seven metrics**, please see Tab. 1 of their paper. Representative baselines of DisEnvisioner, ELITE [2] (ICCV2023 Oral), **only achieved one SoTA metrics across all three metrics**; also the zero-shot (same setting as DisEnvisioner) version of BLIP-Diffusion [3] (NeurIPS 2023), termed BLIP-Diffusion (ZS), **did not achieve any SoTA metrics across all three metrics compared with its baselines**.

[1] PhotoMaker: Customizing Realistic Human Photos via Stacked ID Embedding. CVPR 2024.

[2] Elite: Encoding visual concepts into textual embeddings for customized text-to-image generation. ICCV2023 Oral.

[3] Blip-diffusion: Pre-trained subject representation for controllable text-to-image generation and editing. NeurIPS 2023.

---

### Author Response · Authors · 2024-11-23
**General Response**

We sincerely thank all reviewers for their encouraging and insightful comments regarding the novelty and contributions of our paper, such as “novel approach for disentangled visual prompt processing, holds potential for practical application” (**p8Sz, cZ86, N4Js**), “well-conducted experiments demonstrating the efectiveness with highest overall ranking” (**cZ86, 6rcz**), “impressive visual results” (**N4Js, cZ86**), “good settings of tuning-free and single image-based” (**6rcz, cZ86**), “clear writting” (**N4Js, p8Sz, 6rcz, cZ86**).

We will address all reviewer comments point by point and revise our paper accordingly.

Currently, all revisions are written in the **supplementary material**, here is the added sections:

- **Sec. D: Ablation on CLIP Prior Initialization**
- **Sec. E: Ablation on the Effect of Augmentations in DisVisioner**
- **Sec. F: Ablation on DisVisioner and EnVisioner**
- **Sec. G: Further Ablation on Token Numbers of DisVisioner**
- **Sec. H: Experiments When Using Multiple Reference Images**
- **Sec. I: Quantitative Comparisons with Transformer-based Methods**

We will carefully revise the main paper according to your concerns and our responses.

---

### Meta-Review · Area_Chair_eyuP · 2024-12-17

**Metareview:**

This paper aims to enhance the customization capabilities of image diffusion models. The authors propose a method DisEnvisioner by training the DIsVisioner and EnVisioner modules to address the limitations of existing methods in interpreting subject-essential attributes. Quantitative and qualitative experiments are performed and demonstrate the effectiveness of the proposed method. After rebuttals, all reviewers gave positive rating scores. Based on the above considerations, I recommend accepting this paper.

**Additional Comments On Reviewer Discussion:**

The authors provided rebuttals for each reviewer. Reviewer p8Sz increase the rating to 6 based on the authors’ detailed analysis, Reviewer N4Js and cZ86 provide responses and their concerns are addressed. Based on the consistent opinion of the reviewers, I recommend acceptance as the final decision.

---

### Decision · Program_Chairs · 2025-01-22

Accept (Poster)